# Dissecting Diagnostic and Management Strategies for Plant Viral Diseases: What Next?

B. Megala Devi [1], Samyuktha Guruprasath [2], Pooraniammal Balu [2], Anirudha Chattopadhyay [3], Siva Sudha Thilagar [1], Kanaga Vijayan Dhanabalan [4], Manoj Choudhary [5], Swarnalatha Moparthi [6] and A. Abdul Kader Jailani [5,7,*]

[1] Department of Environmental Biotechnology, Bharathidasan University, Tiruchirappalli 620024, Tamil Nadu, India; megalabiotech@gmail.com (B.M.D.); sudacoli@yahoo.com (S.S.T.)

[2] Department of Chemical and Biotechnology, Sastra Deemed University, Thanjavur 613401, Tamil Nadu, India; 124078008@sastra.ac.in (S.G.); pooraniammal@sastra.ac.in (P.B.)

[3] Pulses Research Station, S.D. Agricultural University, Sardarkrushinagar 385506, Gujarat, India; anirudhbhu@sdau.edu.in

[4] Department of Biological Science, Purdue University, West Lafayette, IN 47907, USA; kvijayan@purdue.edu

[5] Plant Pathology Department, University of Florida, Gainesville, FL 32611, USA; m.choudhary@ufl.edu

[6] Department of Entomology and Plant Pathology, North Carolina State University, Raleigh, NC 27606, USA; smopart@ncsu.edu

[7] Department of Plant Pathology, North Florida Research and Education Center, University of Florida, Quincy, FL 32351, USA

* Correspondence: aamirudeen@ufl.edu

**Abstract:** Recent advancements in molecular biology have revolutionized plant disease diagnosis and management. This review focuses on disease diagnosis through serological techniques, isothermal amplification methods, CRISPR-based approaches, and management strategies using RNA-based methods. Exploring high-throughput sequencing and RNA interference (RNAi) technologies like host-induced gene silencing (HIGS) and spray-induced gene silencing (SIGS), this review delves into their potential. Despite the precision offered by RNAi in pest and pathogen management, challenges such as off-target effects and efficient dsRNA delivery persist. This review discusses the significance of these strategies in preventing aphid-mediated plant virus transmission, emphasizing the crucial role of meticulous dsRNA design for effective viral RNA targeting while minimizing harm to plant RNA. Despite acknowledged challenges, including off-target effects and delivery issues, this review underscores the transformative potential of RNA-based strategies in agriculture. Envisaging reduced pesticide dependency and enhanced productivity, these strategies stand as key players in the future of sustainable agriculture.

**Keywords:** RNA interference; double-stranded RNA; plant disease management; pest control; precision agriculture

## 1. Introduction

In today's agricultural model, plant viruses have become formidable opponents, applying significant economic influence through their devastating impact on plant health [1]. The economic effect of plant viral diseases cannot be overstated, as they cause considerable damage to crops, creating ripples throughout the agricultural supply chain. This affects not only farmers' livelihoods but also food security overall in subsistence agriculture [2]. Furthermore, the economic cost of controlling these viral diseases is of interest, requiring significant investments in pesticides, herbicides, and other disease management strategies [3]. However, this approach also poses challenges to the sustainability of agricultural networks, as the widespread use of chemical agents poses the risk of soil and water pollution, harming non-target organisms and ecosystems [4]. The potential effects of pesticide residues on human health, especially on agricultural workers and consumers, are now under

greater scrutiny [5]. These influences highlight the urgent need to combat plant viruses, and biological stressors that infect crucial crops and contribute to substantial agricultural losses.

In plant virology, the quasispecies concept reveals dynamics within viral populations. Plant viruses, with high mutation rates during replication, generate diverse quasispecies within a host due to the absence of proofreading mechanisms. This structure is adaptive, allowing viruses to respond to diverse conditions and evade host defenses. Genetic diversity within the quasispecies shapes viral evolution and influences the success of infections. Understanding quasispecies dynamics is crucial for developing effective strategies to manage viral infections in agriculture.

Although diseases caused by plant viruses cannot be cured, various management techniques can help prevent and control their spread [6]. Early detection of viruses is crucial to prevent their spread, requiring the adoption of specific and sensitive detection methods. Continuous technological innovations are leading to the development and application of new detection methods in plant virus diagnostics. Conventional detection methods have been time-consuming and unsuitable for viruses with low titers or emerging strains [7]. Therefore, advanced diagnostic methods capable of capturing diverse viral species and their infections are needed. The simultaneous detection of different virus species or strains within the same host plant adds complexity to the diagnosis process. Thus, highlighting the importance of identifying hypervariable regions within the plant virus genome is crucial [8]. Generally, viruses infect plant tissue cells by entering a single cell through wounds or injuries, where they take over the cell's functions to replicate [9]. At that time, viruses may change their genetic makeup either through mutation or genetic recombination to cope with the evolutionary space of host defense [10]. Plant RNA viruses commonly exhibit genetic variation through high mutation rates during replication, forming diverse genetic variants within populations, known as quasispecies. In contrast, DNA plant viruses predominantly evolve through genome recombination or pseudo-recombination. Recombination involves exchanging genetic material between different viral genomes, while pseudo-recombination reshuffles genetic elements within the same viral genome. These mechanisms contribute to the genetic diversity observed in plant virus populations and facilitate their adaptation to changing environments. So that, understanding why specific viruses are highly virulent in different hosts, and implementing sustainable antiviral resistance measures in agricultural settings can be employed [11]. While there are a few instances of naturally occurring genetic resistance that have been incorporated into commercial cultivars, they are relatively rare [12]. Plants lack genetic protection against many viral infections that affect important crops [13]. Fortunately, recent genomic techniques, particularly genome editing and artificial gene silencing, have made it possible to engineer virus resistance in plants and strengthen their immune systems [14,15]. This concept is akin to a vaccine but operates through distinct mechanisms.

For effective plant virus detection, high sensitivity and accurate specificity are necessary. Combinations of multiple techniques are preferred for reliable detection, particularly in situ [16]. Recent developments in molecular biology and genomics techniques have enabled the development of on-spot plant virus diagnostics, allowing for the assessment of plant virus dynamics in the field [17].

This comprehensive review aims to provide a thorough exploration of RNA interference (RNAi) and double-stranded RNA (dsRNA) tools as promising strategies for disease management in plants and pest control. The primary goal is to assess the potential of these molecular tools to revolutionize agriculture by reducing reliance on pesticides, minimizing crop damage, and enhancing overall sustainability. The review focuses on understanding how plants detect viruses and how viruses evade plant defenses, highlighting the role of non-coding RNAs (ncRNAs) in controlling biochemical functions and disease manifestations. Given the significant threat of plant viral diseases to agriculture, the manuscript specifically delves into RNA-based strategies for the diagnosis and management of these adversaries, addressing challenges inherent in plant viral diseases. The exploration includes engineering resistance to plant viruses using various ncRNAs, such as short RNAs and long

ncRNAs, and discusses RNAi-based techniques as potent tools for controlling plant viruses. Additionally, the review covers the comparison between RNAi methods and CRISPR-Cas technologies, as well as the application of RNA-aided CRISPR-Cas gene-editing methods to prevent plant viral infections. The paper also addresses the recent documentation of new plant viruses impacting vegetable production in the USA and covers the two primary methods for engineering virus resistance. It emphasizes the challenges that must be addressed before these technologies can be widely employed to protect crops from viruses [18–23].

## 2. Detection of Plant Viral Diseases

Diagnosis and detection are related concepts but refer to different aspects of identifying a condition, including viral infections. Detection refers to the identification or confirmation of the presence of a particular organism. In the context of viruses, detection involves finding evidence of the virus, such as its genetic material, proteins, or other markers. The primary goal of detection is to establish the existence or occurrence of something. In the case of viral infections, detection could mean identifying the virus itself or its components in a sample, like plant tissue.

In contrast, diagnosis goes a step further than detection and involves the identification and determination of the nature or etiology of a particular condition or disease. In the context of viral infections, diagnosis not only confirms the presence of the virus but also provides information about the specific virus causing the infection. The main purpose of diagnosis is to understand the nature of the condition, its severity, and often its potential implications for treatment and management. Diagnosis may involve additional tests and assessments beyond simple detection. Numerous technologies, such as serological, isothermal, and big data-based approaches, have been employed for the rapid diagnosis of plant viral diseases [24]. All of these methods have been employed to detect various viruses, with PCR being the most widely used technique. RNA isolation from non-model plants is often technically challenging; thus, isothermal techniques are gaining popularity. To date, the high cost and the requirement for laboratory resources have been prohibitive for the mass deployment of these technologies for the real-time, in-field detection of plant viruses.

### 2.1. A Comprehensive Overview of Visualizing Infected Plants and Seeds

Visual symptoms are key indicators of viral diseases in plants, showcasing specific disruptions in plant physiology [25]. These symptoms can manifest in various ways, including mosaic patterns and chlorosis. Mosaic symptoms involve changes in leaf color and shape, while chlorosis is characterized by systemic lesions throughout the plant. Additionally, necrotic lesions can aid in the taxonomic identification of viruses, such as those causing necrotic mosaics in certain crops like cowpea and tobacco [26]. The symptoms of viral diseases include leaf tissue wrinkling, browning, mosaicism, and necrosis [27]. However, diagnosing viral diseases based solely on symptoms is challenging compared to other pathogens. The difficulty arises from the fact that multiple viruses can infect a host and modify symptom manifestations as observed in potato mosaic.

To determine the viral particle concentration in infected plants, researchers employ various methods. One approach involves infecting indicator plants and observing the development of mosaic-like or necrotic spots on their leaves, which vary depending on the viral species [28]. For instance, *Nicotiana glutinosa* L. leaves are used to identify and measure potato virus concentrations, while young tobacco plants (*Nicotiana glutinosa*) serve as indicators for tomato aspermy virus. *Gomphrena globosa* leaves, on the other hand, are utilized in the diagnosis of potato virus x [12]. However, these methods have limitations, as the results can depend on leaf age and the technique used to transfer the virus. Microscopy-based visualization of fluorescently labeled proteins in host plants is considered one of the most effective methods for virus disease diagnosis [25]. Fluorescent proteins are used in viral diagnostics through binding to specific viral proteins. Once the virus infects the host plant, these modified proteins allow scientists to check the virus's behavior and location within plant cells using specialized microscopy techniques [29]. This method supplies a

powerful way to detect viral infections even before obvious symptoms appear, allowing early intervention in agriculture. By supplying real-time information about viral replication and spread, fluorescently labeled proteins will improve our understanding of the complex dynamics between viruses and host cells. In essence, they serve as invaluable molecular markers, shedding light on the world of viral infections in host plants for rapid and accurate diagnosis and research.

Although visual observation serves as a valuable diagnostic tool, allowing for the identification of plant virus infection (Figure 1) and the quantification of virus particles in host plants [30], its authenticity is always subjected to doubt, especially in cases where there are overlapping symptoms or symptoms in fewer plants. Overall, the visual detection technique of plant virus diseases relies on the external manifestations of viral infection in plants, which can also closely resemble specific physiological disorders in plants. This similarity poses challenges in visual-based diagnosis.

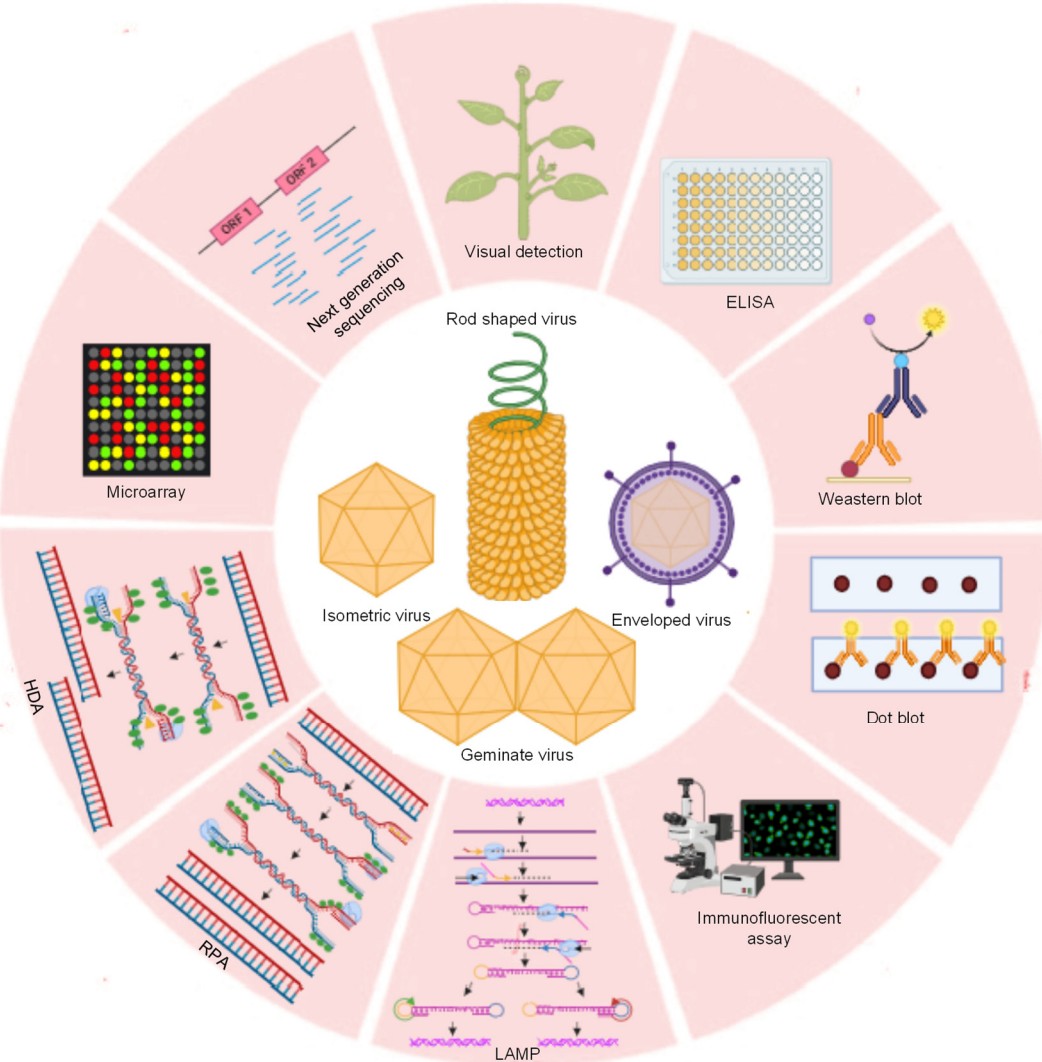

**Figure 1.** Advancement in the detection techniques of plant viruses. Common diagnostic methods used for plant virus identification starting from visual diagnosis to more advanced techniques. Primarily, various serological techniques such as ELISA, Western blot, dot blot, and immuno-fluorescent assay are commonly employed for the detection of plant viruses. Now, techniques have progressed to different isothermal detection strategies like LAMP (loop-mediated isothermal amplification), RPA (recombinase polymerase amplification), and HDA (helicase-dependent amplification). Microarray and next-generation sequencing strategies are now utilized for multiple virus detection and virome analysis in any plant samples.

*2.2. Serological Methods for Detecting Viral Infections in Plants*

2.2.1. Serological Approaches

Serological methods are commonly employed to detect viral infections in plants [31]. These techniques utilize specific antibodies that can recognize and bind to viral proteins or other components of the virus. The binding of these antibodies can be detected using various methods to confirm the presence of the virus in the plant. Some commonly used serological methods for detecting viral infections in plants include:

2.2.2. Enzyme-Linked Immunosorbent Assay (ELISA)

Enzyme-linked immunosorbent assay (ELISA) is an exceedingly sensitive and specific technique used in plant virology to detect both viruses and viral antigens in infected plants. It operates by using antibodies bound to an enzyme, which initiates a reaction upon binding to a specific antigen, resulting in a detectable signal [32]. It is a powerful tool for the detection of plant viruses in general [33]. In practice, ELISA is often employed in various forms, including direct antigen-coated (DAC) ELISA, double antibody sandwich (DAS) ELISA, plate-trapped antigen (PTA) ELISA, etc. Most ELISA modifications have proven effective for plant virus diagnosis, exhibiting varying levels of specificity and accuracy. However, the contemporary use of ELISA has diminished, with other methods being preferred in conditions such as the unavailability of a specific antibody for the target virus. Additionally, the detection limit of ELISA is restricted to 100 pg/mL, which is considerably lower than that of PCR-based molecular techniques. Consequently, ELISA can be effectively integrated with other techniques [34] to enhance the precision of plant virus detection.

2.2.3. Immunofluorescence Assay (IFA)

This method utilizes fluorescent-labeled antibodies capable of binding to viral antigens either directly or indirectly within plant tissues [35]. The bound antibodies can be visualized using fluorescence microscopy. In the case of a direct assay, the primary antibody is fluorescently labeled, while the fluorescently labeled secondary antibody (anti-immunoglobulin antibody) is employed to detect the binding of the unconjugated primary antibody to the antigen. Previously, this technique has been used for the detection of various plant viruses, such as maize chlorotic dwarf virus [36] and potato virus Y [37]. It is crucial to note that specific fluorescent-labeled antibodies are required for each virus, limiting the application of direct immunofluorescence assay in plant virus detection [38]. Furthermore, cross-reactivity and non-specific binding can occur, leading to false-positive results. Moreover, some methods may not be suitable for high-throughput screening, and information about virus strains and variants may not be provided by this method.

2.2.4. Dot Blot Immunoassay

Dot blot assay is a widely used technique for detecting specific viral antigens in plant extracts. In this method, plant extracts, which may have viral antigens of interest are carefully immobilized or "spotted" on the nitrocellulose membrane (NCM). Subsequently, specific antibodies are applied to the membrane. If viral antigens are present in the plant extract, they will bind to specific antibodies, and virus detection can be carried out using an enzyme-linked secondary antibody with the respective substrates. The advantage of DBIA is that a very small amount of sample (2 μL) is required as compared to ELISA (200 μL) [39]. However, it is important to note that Dot Blot testing can have limitations in sensitivity and can sometimes produce false negative results, making this method more suitable for qualitative analysis than quantitative [40]. Accurate detection necessitates relatively concentrated (1 mg/mL) virus antiserum. Furthermore, the higher cost of assay attributed to the expensive NCM and test reagents limits its application. In the past, the application of this method has proven effective in detecting a range of plant viruses, including bean yellow mosaic virus (BYMV), impatiens necrotic spot virus (INSV), and lily symptomless virus (LSV) through dot blot assays [41].

2.2.5. Tissue Blot Immunoassay

Tissue blot immunoassay (TBIA) is a diagnostic technique used to detect the presence of viruses in plant tissues. In this procedure, plant leaves or fruit sap are directly transferred and immobilized to nitrocellulose membranes and then treated with virus-specific antibodies. If a virus is present, the antibody will bind to it, producing a visible colorimetric reaction [42]. TBIA is a sensitive and specific method for detecting plant viruses, with a detection limit of 0.1 ng of viral protein per gram of tissue [43]. This method can diagnose a wide range of viral plant diseases caused by RNA viruses, DNA viruses, and viroids [44].

TBIA offers the advantage of relatively quick performance and requires minimal equipment, making it a useful tool for the rapid diagnosis of viral plant diseases in the field. It is compatible with various plant tissues, including leaves, fruits, and roots [45]. However, TBIA has some limitations. The procedure requires the use of specific antibodies for each virus, which can be expensive and time-consuming to produce [46]. These platforms have effectively facilitated the generation of diverse proteins within various plant systems by harnessing an array of distinct plant viruses. Additionally, TBIA is a qualitative method and does not provide information on viral load or the presence of multiple viral strains [47]. Despite these limitations, TBIA remains a valuable diagnostic tool for viral plant diseases, particularly in situations where rapid diagnosis is crucial for disease management and control (Tables 1 and 2). This technique has effectively identified different plant viruses, including bean yellow mosaic virus (BYMV), impatiens necrotic spot virus (INSV), lily symptomless virus (LSV), TSWV, cucumoviruses, and potyviruses, using tissue blot assays [41,48,49].

**Table 1.** Different types of detection techniques with pros and cons.

| S.No | Detection Technique | Principle | Pros | Cons | Application | Ref. |
|---|---|---|---|---|---|---|
| 1. | Visual Symptoms | Identification of specific disease symptoms | Causes plant physiology abnormalities including chlorosis and mosaic patterns, which help with viral identification and infection tally | A challenge because there are other infections with comparable symptoms. A reliance on virus–host interactions | Ringspot (papaya ringspot virus in papaya), necrosis and wilting (tospoviruses in large numbers of hosts), and swollen shoots (cacoa swollen shoot virus in Cacao) | [50,51] |
| 2. | Indicator Host–Plant Method | Observing mosaic/necrotic spots | To detect the presence of specific plant viruses and for the quantification of viral particles through the formation of local lesions on the inoculated leaves | Applicable for the sap-transmissible plant viruses; sometimes, symptoms development may not be prominent | *Nicotiana tabacum* var. *xanthi* for tomato mosaic virus, *Chenopodium amaranticolor*, *C. album*, and *C. quinoa* for potato virus Y, and *Gomphrena globosa* for potato virus X | [28,52] |
| 3. | Serological Methods | Specific Antigen–Antibody interaction | Precise and sensitive detection; easy for field application | Needs antibodies for every virus. Cross-reactivity's potential to cause false positives. Not necessarily right for high-throughput screening. | ELISA for diverse plant viruses, viz., carmoviruses, potyviruses, tospoviruses, etc. | [53–55] |
| 4. | PCR-based Amplification | Amplification of specific DNA sequences | Highly sensitive and very precise method applicable for genetic testing, forensics, and diagnostics | The need for costly instruments (thermal cyclers), primers, and chemicals (polymerase enzyme) restricts its application to sophisticated laboratories only | Diverse plant viruses (DNA and RNA) for specific gene or genome amplification | [56] |

Table 1. *Cont.*

| S.No | Detection Technique | Principle | Pros | Cons | Application | Ref. |
|---|---|---|---|---|---|---|
| 5. | Isothermal Amplification Assays | Rapid on-site virus detection | The targeted virus genes are amplified using these methods (LAMP, RPA, and HDA) at a consistent temperature, which supplies quicker results, cheaper costs, and a lesser chance of contamination | Some techniques are needed for specialized tools. For precision, primer design is critical. Accuracy and sensitivity are enhanced when combined with CRISPR/Cas | LAMP, RPA, and HDA for the on-site detection of multiple plant viruses | [57–60] |
| 6. | CRISPR/Cas-based Detection (CRISPR-Dx) | Coupling with isothermal assays | Combines CRISPR/Cas detection and isothermal amplification. Rapid, sensitive, focused, and simple to work. Applied successfully for several plant viruses | Requires familiarity with the target viral sequences. For some viruses, optimization might be needed. Not as commonly used as other techniques | Detection of tomato yellow leaf curl virus and potato virus Y | [61,62] |
| 7. | CRISPR-based Approach | Detection using CRISPR-Cas12a | Fluorescence signal-based detection that is quick and correct. No need for amplification or reverse transcription. Cheaper and more efficient procedure | Requires familiarity with the target viral sequences. For some viruses, optimization could be needed. Not as commonly used as other techniques | Agricultural plant virus detection | [63] |
| 8. | Microarray in Plant Viral Disease | Simultaneous detection of viruses | Substantial specificity and sensitivity. Quick and efficient throughput. Cost-effective. Identification of several viral pathogens in-depth | Needs unique probes for every virus. Restricted to recognized viral strains. The need for special tools for detecting fluorescent markers | Identification of plant viruses and strains of multiple genera (obamovirus, tobravirus, tospovirus, potexvirus, potyvirus, calavirus, comovirus, cucumovirus, ilarvirus, nepovirus, pomovirus, sobemovirus, etc.) | [64–66] |
| 9. | Metagenomics in Plant Viral Disease | Analysis of the virome within infected plant samples | Detection and characterization of viral communities within plant samples, so that new viral strains or variations can be found | Requirement for innovative sequencing technologies and analysis of complex data is an issue | Identification of multiple viral pathogens or novel plant virus species, within the families chrysoviridae, endornaviridae, partitiviridae, totiviridae, etc. | [67–70] |
| 10. | High-throughput Sequencing | Detection of multiple virus species | To understand viral diversity, evolution, and interactions with host plants | Difficulty in big data analysis and challenges in the accurate assemble and annotation of novel plant viruses | Detection and characterization of viruses in citrus, grapevine, etc. | [71–73] |
| 11. | Tissue Blot Immunoassay | Detection using virus-specific antibodies | A method for detecting viruses that is sensitive and precise. Effortless performance with little equipment needed | Demands unique antibodies for every virus. A qualitative approach without considering viral load or strain diversity | Rapid diagnosis of plant viruses like citrus tristeza virus, tomato spotted wilt virus, etc. | [74] |

**Table 2.** Various techniques employed for plant virus detection and diagnosis.

| Disease | Plants Affected | Pathogen | Effects of the Disease | Diagnostic Techniques | Related Articles |
|---|---|---|---|---|---|
| Cassava brown steak disease (CBSD) | Cassava (*Manihot esculenta*) | Cassava brown steak virus (CBSV) and Ugandan cassava brown steak virus (UCBSV) | Necrosis, root rottening, and stunted growth | Visual inspection, molecular assays, and HTS | [75] |
| Maize chlorotic mottle virus (MCMV) | Maize (*Zea mays*) | Tombusviridae | Mottling on the leaves | ELISA | [76,77] |
| Banana bunchy top disease | Banana (*Musa acuminata* and *Musa balbisiana*) and plantain Banana | Banana bunchy top virus (genus Babuvirus) | Small deformed fruit/no fruit | LAMP | [78] |
| Citrus tristeza disease | Citrus | Citrus tristeza virus (genus Closterovirus) | Twig dieback and browning of bark | RT-PCR | [79] |
| Potato tuber necrotic ringspot disease | Potato (*Solanum tuberosum*) | Potato virus Y strains NTN (genus Potyvirus) | Brown rings on the surface and yelloweye | Western blot | [80] |
| Tomato spotted wilt disease | Tomato (*Solanum lycopersicum* L.) | Tomato spotted wilt virus (genus orthotospovirus) | Stunting, necrosis, chlorosis, and ring spots | Cas 13a | [81] |
| Zucchini yellow mosaic disease | Cucurbits | Zucchini yellow mosaic virus (genus, potyvirus) | Reduced leaf lamina and necrosis | RT-PCR | [82] |
| Papaya ringspot disease | Papaya (*Carica papaya*) | Papaya ringspot virus (genus Potyvirus) | Water-soaked oily streaks | TBIA | [83] |
| Carrot virus Y | Carrot (*Daucus carot*) | Carrot virus Y | Chlorotic mottle | ELISA | [84] |
| Subterranean clover stunt disease | Legumes | Subterranean clover stunt virus (SCSV) | Cupping of leaflets | RT-PCR | [85] |

Note: RT-PCR (reverse transcriptase-polymerase chain reaction), TBIA (tissue blot immunoassay), LAMP (Loop-mediated isothermal amplification), ELISA (enzyme-linked immunosorbent assay), and HTS (high-throughput sequencing).

2.2.6. Western Blotting

Western blotting is a technique used to detect a specific protein in a biological sample (containing complex proteins) via separation through electrophoresis. This technique involves separating viral proteins and transferring them onto a membrane, followed by binding with a specific antibody. Specific antibodies are then used to detect the viral proteins on the membrane [86]. Western blotting (WB) can effectively address the issues associated with ELISA, particularly in reducing the number of false positives. Thus, this technique has been successfully applied in southern rice black-streaked dwarf virus and wheat streak mosaic virus (WSMV) [87].

Overall, these comprehensive serological approaches have limited field application for plant virus detection, but they are routine techniques for the laboratory diagnosis of plant viruses. In general, serological methods, mainly ELISA, play an important role in the reliable detection of viral infections in plants. The accuracy of virus detection can be improved when supplemented with other methods such as DNA and RNA amplification followed by sequencing. These advanced molecular techniques not only confirm the presence of viruses but also identify the specific virus or virus strain in question. Thus, the integration of advanced molecular techniques, specifically DNA and RNA amplification followed by sequencing, has significantly elevated the accuracy and specificity of virus detection in the realm of plant virology. A noteworthy example lies in the use of reverse transcription polymerase chain reaction (RT-PCR) followed by sequencing, which has proven instrumental in identifying and characterizing various strains of RNA viruses responsible for mosaic diseases in crops. Furthermore, the advent of next-generation sequencing (NGS) technologies has ushered in a new era, enabling high-throughput sequencing of entire viral genomes in plant viruses. NGS has been particularly effective in unveiling novel viruses and understanding the intricate diversity within viral populations. Loop-mediated isothermal amplification (LAMP), a technique facilitating isothermal DNA amplification, when complemented with sequencing, has demonstrated enhanced precision in identifying specific viral sequences in plant viruses, as exemplified in the detection and characterization of RNA viruses like tomato spotted wilt virus. These examples underscore the transformative impact of combining amplification and sequencing methodologies, providing researchers with powerful tools to unravel the complexities of viral infections in plant species. Serological techniques can be effectively integrated with other techniques for the comprehensive detection of virus infection in plants.

*2.3. PCR-Based Thermal Amplification for Plant Virus Detection*

The development of PCR (polymerase chain reaction) in 1984 marked the next significant advance in the identification of viruses. By amplifying a particular portion of the virus genome for detection through sequencing or fingerprinting, the approach increases the assay's sensitivity several times over when compared to other serological assays like ELISA. A few common processes in conventional PCR are extracting nucleic acids from the test plant, designing primers unique to the virus, and assembling the PCR in a vial with the addition of magnesium chloride, Taq polymerase, primers, and nucleotides. After that, the vial is put into the heat cycler with pre-set settings for denaturation, primer annealing, and extension. Finally, detection can be carried out after running the contents of the vial either on an agarose gel (in fingerprinting) or sequencer (in sequencing). So far, many PCR variations have been created, including multiplex PCR, nested PCR, immunocapture PCR, reverse transcription PCR (RT-PCR), real-time PCR, and more. Of them, nested PCR is a modification of the conventional PCR technique that involves two sets of primers to amplify a specific DNA fragment. The method is particularly useful when working with DNA samples of low concentration or when trying to detect a target sequence in the presence of closely related sequences. Furthermore, multiplex PCR is used for the simultaneous detection of several viruses infecting a sample. Later, real-time PCR (qPCR) was developed to measure the accumulation of PCR products in real-time using fluorescent dyes or probes; thus, it can be used to detect the virus titer in a sample with high sensitivity and specificity. Since even a few copies of the viral nucleic acid contained in the test sample can be amplified and detected, qPCR is highly preferred for plant virus detection as well as quantification. Similarly, digital PCR is a more recent advancement in PCR technology that allows for absolute quantification of nucleic acids. It can be particularly useful in precise viral load determination. The choice of PCR-based thermal amplification method depends on the specific goals of the experiment, the nature of the target nucleic acid, and the available resources. Each technique has its own advantages and limitations, and researchers can select the most suitable method based on the experimental design and objectives for monitoring the plant samples.

### 2.4. Utilizing Isothermal Amplification Assays for On-Site Plant Virus Detection

Isothermal amplification assays have emerged as powerful tools for the rapid and on-site detection of plant viruses. These assays enable the amplification of specific regions of the viral genome at a constant temperature, eliminating the need for thermal cycling. Some commonly used isothermal amplification assays for on-site plant virus detection include:

### 2.4.1. Loop-Mediated Isothermal Amplification (LAMP)

LAMP is a simple and robust method designed based on the DNA strand displacement activity of *Bst* DNA polymerase. It utilizes four to six primers targeting multiple regions of the viral genome for isothermal amplification [88]. Out of four primers, two inner primers consisting of forward inner primer (FIP) and backward inner primer (BIP) sequences recognized a sense and an antisense sequence of the target DNA, whereas two outer primers including forward outer primer (F3) and backward outer primer (B3) recognized only one external sequence of the target DNA [89]. Isothermal amplification is carried out at 60–65 °C, the optimum temperature for Bst polymerase activity. There are several ways to identify its products: colorimetric (SYBR Green and hydroxyl naphthol blue–HNB) visualization, precipitate identification (turbidity of magnesium pyrophosphate), fingerprinting display (using agarose gel electrophoresis), and real-time detection (using intercalating fluorescent dyes) [90–92]. It can detect viral DNA or RNA and has demonstrated high specificity and sensitivity for plant virus detection. However, LAMP may require specialized equipment for amplification and detection, and optimizing primer design is crucial for accurate and specific detection [93]. Comparable to qPCR, LAMP is substantially faster (45–60 min), and approximately 10–100 times more sensitive than PCR [94]. Thus, LAMP is a versatile tool with great potential to be deployed on-site. LAMP has been employed for the detection of numerous plant viruses. Some examples of LAMP-based plant virus detection include tobacco mosaic virus (TMV), tomato leaf curl new Delhi virus (ToLCNDV), tomato spotted wilt virus (TSWV), watermelon mosaic virus, and zucchini yellow mosaic virus [95–99].

### 2.4.2. Recombinase Polymerase Amplification (RPA)

RPA is another isothermal amplification technique which requires the recombinase enzyme, single-stranded binding (SSB) proteins, and the polymerase enzyme to amplify the target DNA or RNA at the ambient temperature/constant temperature close to 37 °C. It is a very rapid and sensitive method, similar to LAMP but has lower efficiency. This is a relatively new technique and is extensively used nowadays for the detection of many plant viruses including little cherry virus 2 (LChV2), yam mosaic virus (YMV), plum pox virus (PPV), begomoviruses such as bean golden yellow mosaic virus (BGYMV), tomato mottle virus (ToMoV), and tomato yellow leaf curl virus (TYLCV), and banana bunchy top virus (BBTV) [100–106] [Figure 2A]. During the RPA reaction, the recombinase enzyme combines with the primers to form a recombinase-primer complex. In the presence of ATP and a crowding agent (polyethylene glycol) [107], this complex recognizes the complementary sequence on the template. After the recombinase separates the dsDNA strands, primer annealing takes place in the open loop, leaving the 3′ end available to the DNA polymerase to extend the chain, and the single-stranded binding protein (SSB) stabilizes the separated DNA strands. As the DNA polymerase extends the chain, a new dsDNA synthesis event occurs. The end-point product can be identified using fluorescein on one side and a biotin that is attached to the reverse primer/probe or antibody-based lateral flow devices.

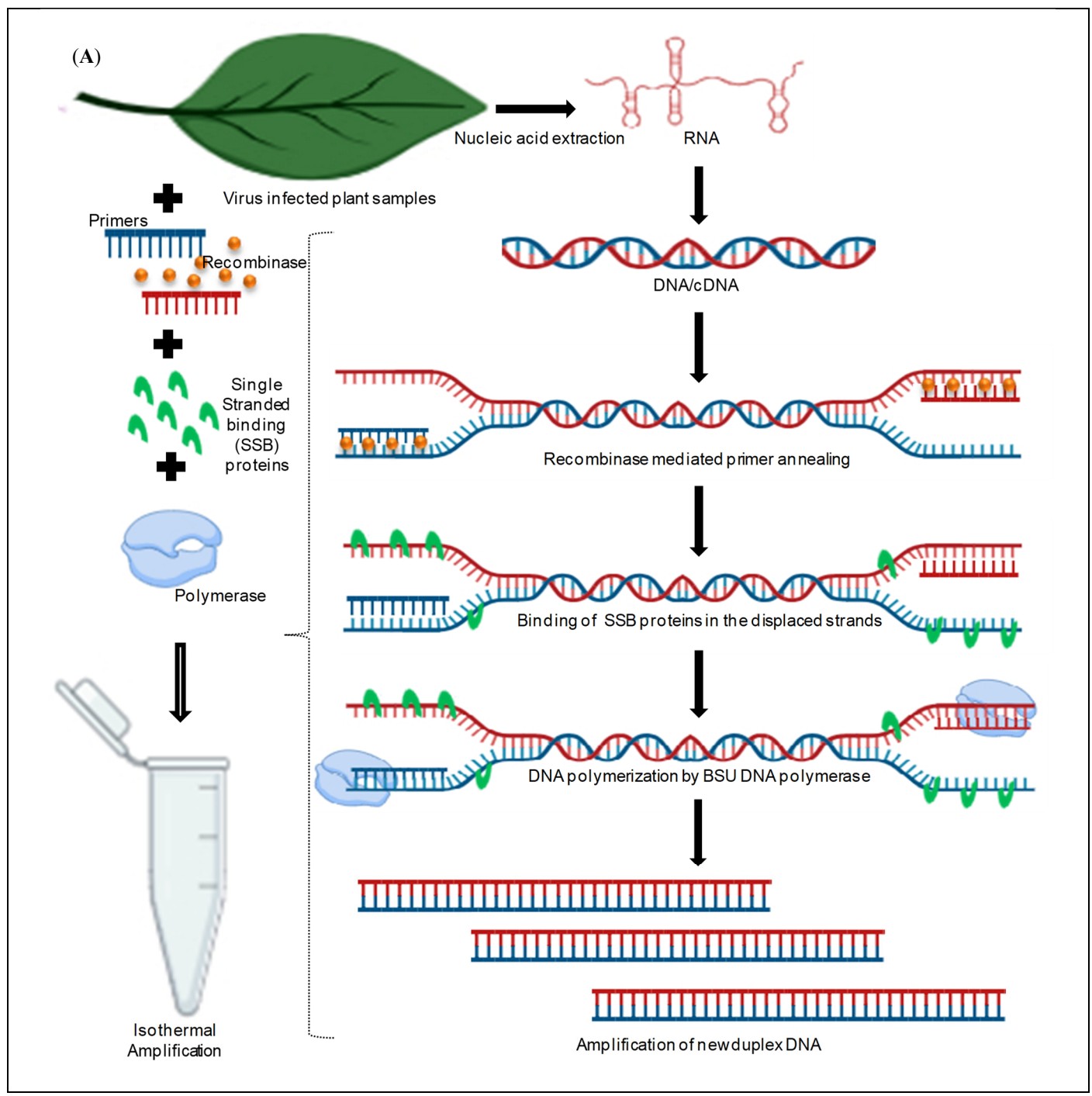

**Figure 2.** *Cont*.

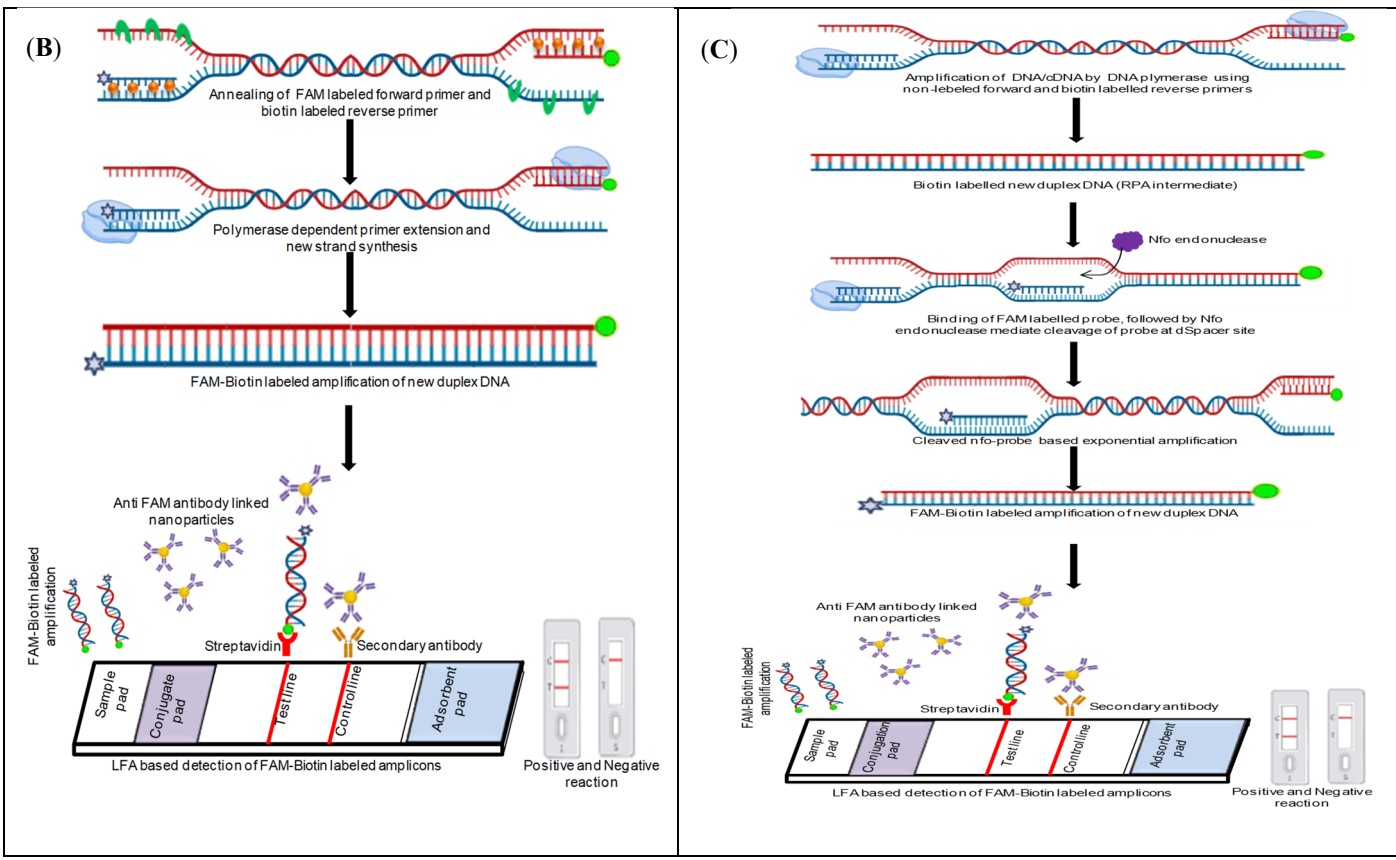

**Figure 2.** Recombinase polymerase amplification (RPA)-based detection of plant viruses. (**A**) Extracted nucleic acids from the infected plant samples are subjected to treatment with recombinase, single stranded binding (SSB) proteins and polymerase, which leads to its isothermal amplification. First, recombinase proteins assemble complexes with both the forward and reverse primers in order to search the DNA for similar sequences. The strand-displacement activity of the recombinase then inserts the primers at the appropriate position, and SSB proteins stabilize the displaced DNA single strands. After the recombinase breaks down, a DNA polymerase can access the primer annealed with the DNA strand and extend it. By repeating this process in cycles, exponential amplification is obtained. Detection of RPA products in LFA using nuclease-independent labeled probes (**B**) and nuclease-dependent labeled probes (**C**). In the nuclease-independent strategy (**B**), the FAM-labeled forward primer and the biotin-labeled reverse primer were used for RPA, and FAM–biotin-labeled amplicons were detected in the lateral flow assay using anti-FAM antibody-linked nanoparticles and streptavidin. Amplicons containing biotin were captured and produced a signal at the test line and unbound nanoparticles labeled with anti-FAM antibodies (primary) were captured at the control line, with them containing the secondary antibody.

### 2.4.3. Helicase-Dependent Amplification (HDA)

HDA is another isothermal amplification method that utilizes helicase enzymes to unwind double-stranded DNA and primers to amplify the target sequence within the genome with the help of DNA polymerase [108]. The simultaneous chain reaction continued for the exponential amplification of the target sequence. HDA has demonstrated sensitivity and specificity for detecting plant viruses [109]. It has several advantages over other isothermal amplification methods like LAMP, RPA, etc., by having a simple isothermal reaction process at one temperature. But the problem of non-specific amplification in HAD is the main limiting factor and restricts its widespread application. This method was previously utilized for the detection of tomato spotted wilt virus through helicase-dependent DNA amplification [110].

These isothermal amplification assays can be performed using portable instruments such as the Genie III or the Biomeme Franklin, enabling the on-site detection of plant viruses [111]. They offer several advantages over traditional methods, including faster results, lower cost, and reduced risk of contamination. These assays are particularly useful for monitoring and controlling the spread of plant viruses in field settings.

Isothermal amplification assays, such as LAMP or RPA, can also be combined with CRISPR/Cas technology for the detection of plant viruses [112]. This approach, known as CRISPR/Cas-based detection (CRISPR-Dx), harnesses the programmable nature of the CRISPR/Cas system to specifically target and cut viral RNA or DNA sequences. By coupling isothermal amplification with CRISPR/Cas detection, the amplified product is mixed with a CRISPR/Cas system designed to recognize the viral RNA or DNA. Upon detection, the CRISPR/Cas system triggers its collateral activity, resulting in the cleavage of a reporter molecule and generating a visual signal, indicating the presence of the virus [113] (Figure 2B). This approach offers advantages in terms of speed, sensitivity, specificity, and ease of use. It has been successfully applied for the detection of various plant viruses, including tomato mosaic virus (ToMV) and tomato brown rugose fruit virus (ToBRFV) [114].

### 2.5. CRISPR-Based Approach for Plant Virus Detection

Cas12 is primarily recognized for its precise targeting of DNA; advancements in CRISPR technology have enabled its application for the detection of viral RNA. In this diagnostic context, Cas12 can simplify the identification of viral RNA, eliminating the need for certain complex steps such as reverse transcription or amplification. This adaptability showcases the versatility of CRISPR-Cas12 in addressing specific diagnostic requirements beyond its natural DNA-targeting function [115]. It also cleaves non-target single-stranded DNA, thereby improving detection. The resulting fluorescence signal is rapidly detected with minimal equipment and ability. The high sensitivity of this method allows it to detect even small viral infections, making it invaluable in detecting plant viruses, especially in regions with limited resources.

The detection process involves several steps: first, the Cas12a enzyme is programmed to recognize particular viral RNA or DNA sequences. When these sequences are encountered in a plant sample, Cas12a cleaves and activates the viral nucleic acid. It also cleaves non-targeted single-stranded DNA molecules, contributing to the detection reaction. This cleavage event releases a fluorescent signal that can be identified using a fluorescent reader [116]. The presence of this signal confirms the presence of the target virus in the plant sample. The method enables rapid and highly sensitive detection, making it valuable for agricultural plant virus detection [112].

A significant advantage of this method is its ability to detect viral RNA without requiring reverse transcription or amplification, which are necessary in many other detection techniques like RT-PCR [8]. This eliminates the need for complex laboratory equipment and expertise, making the process simpler, faster, and more cost-effective [117]. Overall, CRISPR-Cas12-based detection of plant viruses offers a powerful and promising approach for rapidly and accurately detecting viral pathogens in plants. Its simplicity, sensitivity, and specificity make it particularly suitable for use in remote areas with limited access to experimental facilities [118]. Another advantage of CRISPR-Cas12-based detection methods is that they do not require additional amplification steps such as reverse transcription or polymerase chain reaction (PCR), commonly used in other diagnostic techniques. This streamlines the process, reduces costs, and minimizes the risk of contamination [119]. Furthermore, CRISPR-Cas12-based detection methods have demonstrated high levels of specificity and sensitivity, capable of detecting even very small viral infections in plant samples, with detection limits as low as a few copies of viral RNA per microliter of sample [112]. CRISPR-Cas12 offers cost-effectiveness, high specificity, and sensitivity in detecting plant viruses. It is affordable, minimizes false positives, and detects small viral infections, making it ideal for the early and correct detection of pathogens in remote, resource-limited areas. In summary, the utilization of CRISPR-Cas12 for field-scale plant virus detection holds great

promise as an effective approach for the early and accurate detection of viral pathogens. This technology facilitates improved disease control and higher crop yields [112]. A previous study presents the creation of an in vitro one-step CRISPR-based assay for nucleic acid detection, termed iSCAN-OP, designed for the diagnosis of potato virus X (PVX) and tobacco mosaic virus (TMV) [120].

### 2.6. Exploring Big Data Methods

2.6.1. Microarrays in Plant Viral Disease Diagnosis

DNA microarrays are unbelievably valuable tools for diagnosing viral diseases in plants because they allow for simultaneous detection of multiple viral pathogens in a single assay. This high-throughput screening method allows for the rapid and correct identification of plant viruses and their strains [121]. A microchip involves the use of specialized glass panels called microchips. These chips are covered with many small spots of synthetic DNA or RNA probes that match the specific viral sequence [122]. To detect viral pathogens, an infected sample is placed on a DNA chip, and the viral RNA in the sample binds or hybridizes with complementary DNA or RNA probes on the chip surface [123]. This binding is then detected using a fluorescent marker, allowing for visualization and quantification of viral sequences [124]. Using DNA microarrays to diagnose viral diseases in plants offers several advantages over traditional methods, including: (1) high sensitivity: microarrays can detect viral RNA even at low levels, making them overly sensitive diagnostic tools. They specifically recognize target viral sequences, reducing the risk of false positives [125]. (2) Fast and throughput: microarrays can process large numbers of samples simultaneously, supplying fast and efficient results [126]. (3) Cost-effectiveness: DNA chips are relatively inexpensive to produce and can be used for several tests [127]. (4) Comprehensive detection: a single microarray assay can detect multiple viral pathogens, supplying a comprehensive view of the viral pathogens present in a sample. In summary, DNA microarrays are powerful tools for diagnosing viral diseases in plants due to their ability to simultaneously detect multiple viral pathogens [128]. They are used in combination with specialized glass slides called microarray chips and offer advantages such as increased sensitivity, specificity, speed, cost-effectiveness, and comprehensive detection [129]. The utilization of a microarray-based technique has proven to be effective in identifying specific plant viruses. Notably, it has been used to successfully detect the presence of cucumber mosaic virus, potato virus Y, and potato leaf roll virus (PLRV) [130].

2.6.2. Metagenomics in Plant Viral Disease Diagnosis

Metagenomics is a branch of molecular biology that involves analyzing genetic material directly from environmental samples, such as soil and water [131]. In the context of plant viral diseases, metagenomics can be employed to identify and characterize viruses present in plant samples and study their potential interactions with other microorganisms [132]. Metagenomics provides a more comprehensive understanding of plant viral diseases by enabling the detection of multiple viral pathogens in a single sample, including those that may have previously gone undetected. It also aids in the identification of new virus strains or variants that may arise within plant populations [133].

The process of metagenomic analysis typically involves extracting genetic material from plant samples, sequencing it using next-generation sequencing technologies, and utilizing computational methods to analyze the resulting data [134]. This analysis provides information about the diversity of viral pathogens (virome) present in the sample, their genetic composition, and their potential interactions with host plants and other microbes [135]. Although each method has significant drawbacks, when combined, these findings highlight how little we actually know about plant viruses and highlight the need for more thorough research. Metagenomics, coupled with high-throughput sequencing, has showcased its effectiveness in diverse applications, contributing significantly to our comprehension of plant viral diseases and offering valuable insights for the development of more efficient disease management and control strategies.

2.6.3. High-Throughput Sequencing in Plant Viral Diseases

High-throughput sequencing (HTS) has revolutionized research on plant virus diseases by enabling the detection and identification of multiple virus species in a single sample. HTS technology enables the rapid generation of large amounts of sequencing data, which can be used to identify the presence of viral pathogens, track their spread and evolution, and develop new strategies for disease control [136].

One major advantage of HTS in plant virology is its capability to detect newly emerging viruses. Traditional diagnostic methods, such as serological and molecular assays, are designed to detect specific viruses and are often limited to known pathogens [137]. In contrast, HTS allows for the detection of new viruses and the discovery of new strains or variants of known viruses. HTS has also been instrumental in studying the complex interactions between viruses and their host plants [138]. For example, transcriptomics and metagenomics can be employed to study changes in gene expression and the composition of the plant virome during infection, providing insights into viral pathogenesis mechanisms and potential targets for disease control [139].

Another area where HTS has proven particularly useful is in the development of new disease management strategies. HTS data can be used to design virus-resistant crops or develop novel antiviral agents [140]. Furthermore, HTS can be utilized to monitor the effectiveness of disease control measures, such as vaccination and chemotherapy [141]. The utility of HTS in disease management is its application in tracking the effectiveness of integrated pest management (IPM) strategies. HTS data can be employed to assess the impact of biological control agents, cultural practices, and other IPM measures in mitigating plant viral diseases, providing valuable insights for optimizing sustainable and environmentally friendly disease control approaches. Overall, HTS has transformed the field of plant virology by providing powerful tools for the detection, identification, and characterization of viral pathogens. As HTS technology continues to advance, it is likely to play an even more crucial role in developing new strategies for disease control and management.

*2.7. Application of Diagnostic Tools in Agriculture*

Diagnostic tools play a crucial role in agriculture and plant disease management. These tools help identify and detect plant pathogens, enabling timely and targeted responses to prevent the spread of diseases. Here are some key applications of plant pathogen diagnostic tools:

1.  Early detection of the pathogens for disease monitoring and surveillance: Early detection allows for prompt action to control the spread of the disease before it causes significant damage. Various PCR (polymerase chain reaction)- and qPCR (quantitative PCR)-based techniques can be employed to detect specific DNA or RNA sequences of plant pathogens. Nowadays, LAMP- and RPA-based isothermal techniques are preferred as the rapid and cost-effective method for the in situ detection of plant pathogens.

2.  Quarantine and Trade Regulation—Molecular Barcoding: Using molecular markers to identify and track specific plant pathogens helps in enforcing quarantine measures and regulating the international trade of plants and plant products to prevent the introduction of harmful pathogens into new regions. Phytosanitary measures are crucial to prevent the introduction and spread of plant pests and diseases that can have significant economic and environmental impacts. Detection and diagnostic tools play a key role in implementing effective phytosanitary measures. Here are some important tools in this regard. Effective phytosanitary measures often involve a combination of these tools and a collaborative approach between governments, agricultural authorities, researchers, and farmers to ensure the prevention and control of plant pests and diseases. Regular training programs for those involved in monitoring and diagnostics are also essential to enhance the efficacy of these measures.

3. Decision Support Systems: Diagnostic tools contribute to IPM by providing data that can be integrated into decision support systems. This enables farmers to make informed decisions on disease management strategies.

4. Integrated Disease Management (IDM): By employing these diagnostic tools, farmers and plant health professionals can make informed decisions to manage and mitigate the impact of plant diseases on crop yields and agricultural productivity.

## 3. Strategies for the Targeted Management of Plant Viruses

Management options for plant viral infections in the field are currently limited; intracellular pathogens such as viruses are impervious to chemical control measures. Additionally, the utilization of pesticides and other hazardous agents serves to hinder the virus's transmission through vectors. However, these are necessary but undesired in the context of environmental safety. Because pesticides are often used extensively, insect vectors have developed resistance mechanisms against them. Furthermore, the possibility of the emergence of very aggressive viral species and strains through mutation and genome recombination leads to overcoming genetically imposed resistance in plants. Therefore, the need of the hour is to identify potential options with enhanced efficacy and environmental safety. A combination of diverse management strategies can be employed against plant viruses and their vectors with the activation of immune reactions in plants either through RNA interference (RNAi) or via CRISPR interference (CRISPRi). Activation of specific RNAi- and CRISPRi-mediated defense pathways in virus-infected plants is of great interest. When compared to chemical pesticides and transgenic approaches, these interferences can be employed to suppress viral activity and inhibit the virus transmission capacity of insect vectors, even though several researchers have stated the importance of integrated management options to protect plants from plant viral diseases.

### 3.1. RNA Interference for Plant Virus Management

RNA interference (RNAi) is a biological process in which the host molecular RNA inhibits gene expression by neutralizing the specific target mRNAs of pathogens [142]. RNAi has emerged as a promising tool for targeted gene regulation in plants and has been used in the management of plant diseases [143]. RNA plays a crucial role in plant disease detection as it contains essential information about the presence and activity of pathogens. In the context of plant diseases, RNA serves as a molecular marker reflecting the genetic material of pathogens, including viruses and other infectious agents [144]. By detecting specific RNA molecules, such as viral RNA, researchers can identify the presence of pathogens and monitor their activity in plant tissues. RNAi can be employed to control plant diseases by suppressing the expression of genes involved in the pathogenicity of the disease-causing agent. This approach is achieved by introducing small interfering RNAs (siRNAs) into plant cells, which specifically target and destroy the pathogen's mRNA molecules. Consequently, the expression of the pathogen's virulence genes is reduced, leading to a decrease in disease symptoms [145]. Another application of RNAi in disease control is targeting susceptible genes in the plant that participate in the disease response pathway. By suppressing these susceptible genes, RNAi prevents the plant from mounting an excessive immune response that can result in the damage and death of plant cells [146,147].

RNAi-based approaches for disease control offer several advantages over traditional chemical pesticides. RNAi is highly specific, exclusively targeting the genes of the pathogen without affecting non-target organisms. Additionally, RNAi is environmentally safe, leaving no chemical residues that can accumulate over time. Overall, RNAi has tremendous potential for the targeted management of plant diseases. Through ongoing research and development, RNAi-based approaches could become a crucial tool for sustainable agriculture and crop protection [148].

Small interfering RNA (siRNA) duplexes, 19–25 nucleotide double-stranded RNA molecules created through DICER-mediated cleavage of larger double-stranded RNAs

(dsRNA), are responsible for initiating RNAi. The corresponding mRNA is then cleaved endonucleolytically due to the addition of a single-stranded guide RNA to the RNA-induced silencing complex (RISC), which controls the expression of the targeted gene [149]. While perfect similarity between the mRNA and the entire siRNA sequence is not necessary, a region of approximately 8 base pairs, known as the seed sequence, is sufficient for RNAi-mediated silencing to occur, making it a distinct mechanism [150]. Any mRNA with a perfect base complementarity to the gene encoding the guide strand can be downregulated by RISC [151]. Cellular RNA-dependent RNA polymerases (RdRPs) may enhance the RNAi effect through the transitive creation of secondary siRNAs by amplifying the antisense strand of the mRNA target. The gene silencing response is triggered by exposure to foreign genetic material, often dsRNA [152]. MicroRNAs (miRNAs), which are short RNAs (sRNAs) spontaneously produced from specific genome-encoded precursors, mediate host immune systems [153], pathogen pathogenicity [154], and host–pathogen communication [155], in addition to defending organisms against foreign nucleic acids. Prokaryotes lack RNAi machinery, but they possess a defensive mechanism that inactivates parasite genomes, which functions similarly: they generate short non-coding RNAs that can up- or downregulate mRNA stability and translation [156].

Certain RNAi procedures can fail due to the potential inadvertent silencing of non-target genes, which can have an impact on plant physiology. Achieving efficient delivery of double-stranded RNA (dsRNA) into plant cells can be a challenging task that may necessitate the use of specialized techniques. Additionally, dsRNA is susceptible to degradation in the environment, which limits its long-term effectiveness. It is important to note that the efficiency of RNAi can vary among different virus strains and mutants, and some viruses may even develop resistance to RNAi. These factors contribute to the complexity and variability associated with RNAi-based approaches in plant disease management. Ongoing research and advancements in delivery methods and understanding of virus–host interactions are crucial for addressing these challenges and enhancing the overall effectiveness of RNAi in controlling plant diseases [157,158] (Table 3).

**Table 3.** RNAi-based strategies for the management of plant viral diseases.

| S.No | Strategies | Target | Pros | Cons | Example | Reference |
|------|-----------|--------|------|------|---------|-----------|
| 1 | RNA Interference (RNAi) | Targeted gene regulation | Specific pathogen gene targeting without hurting species that are not the target. Sustainable and safe for the environment. Possibility of protecting crops and controlling illness. | Possibility of unintentionally silencing non-target genes. Difficulties with dsRNA delivery and environmental deterioration. Efficacy varies across viral strains. | Targeted management of plant diseases. | [152–155] |
| 2 | Nanoparticle-encapsulated dsRNA | Selective gene silencing in insects | Reduction in pest populations without endangering creatures is not the goal. Broad-spectrum pesticides must be eliminated. | Delivery problems for dsRNA and safety optimization. Little data from real-world applications. | Eco-friendly substitute for traditional pest control techniques. | [159] |
| 3 | RNAi in Pest and Pathogen Control | Use of HIGS, SIGS, AND VIGS | Transgenic plants that express dsRNAs tailored to certain pests or pathogens. Goods that are offered for sale. | Difficulties with dsRNA specificity and design. The efficiency of transformation varies. | Targeted management of pests and pathogens. | [160] |

**Table 3.** *Cont.*

| S.No | Strategies | Target | Pros | Cons | Example | Reference |
|------|-----------|--------|------|------|---------|-----------|
| 4 | Topical Application of dsRNA against Aphids | RNAi-mediated inhibition of virus transmission | DsRNAs consumed by aphids cause RNAi, which destroys viral RNA. | A specific dsRNA design is needed. Promising results in stopping viral transmission caused by aphids. | A low-cost means of preventing viral infections caused by aphids in crops. | [161] |
| 5 | Nucleic Acid Extraction and dsRNA Generation | Isolation and purification of dsRNA | dsRNA production for research on plant–virus interactions and gene silencing. | Choosing the best techniques for dsRNA characterization and synthesis. | dsRNA production is a fundamental technology for numerous applications. | [162–164] |
| 6 | Impact of dsRNA on Cucurbitaceae Virus Control | Use of dsRNA in gene silencing | Numerous techniques for producing dsRNA for investigations on plant–virus interactions and gene silencing. | Plant sensitivity to agro-inoculation varies. | Recognizing dsRNA's potential for preventing viruses in the Cucurbitaceae family. | [165] |
| 7 | dsRNA in Aphid-Mediated Virus Control | RNAi-induced control of viral transmission | DsRNAs administered exogenously prevent viral transmission carried out by aphids. | Require a certain dsRNA design. Effectiveness in halting the spread of viruses. | A promising method for managing aphid populations and avoiding the transmission of viruses that are transmitted by them. | [166] |

### 3.2. CRISPR Interference (CRISPRi)

The CRISPR (clustered regularly interspaced short palindromic repeats)/Cas system is used to introduce precise changes in the plant's DNA. This can involve deleting, adding, or modifying specific DNA sequences. This technology has been applied to develop virus-resistant plants through the precise editing of either the plant's genome or viral genome. This approach offers an efficient way to obtain plant resistance to viral infections through various strategies viz.

1. Modification of Host Factors: Some strategies involve modifying host factors that are essential for viral replication. By disrupting or modifying these susceptibility factors, researchers can hinder the virus's ability to complete its life cycle within the plant. Several examples demonstrate the modification of host factors using CRISPR technology to enhance plant virus resistance. One such notable example is CRISPR-edited *Nicotiana benthamiana* for enhanced immunity against plant viruses. Scientists used CRISPR to edit UbEF1B and CCR4/NOT3, which are crucial for the replication of geminiviruses. The edited plants exhibited increased resistance to specific plant viruses [167]. Similarly, the editing of the translation initiation factor, the eIF4E1 gene in tomato plants, leads to enhanced resistance to infection by certain RNA viruses, including the pepper mottle virus [168]. Furthermore, the targeted modification of susceptibility (S) genes (eIF4G, Vacuolar ATPases subunit D) in rice showed enhanced resistance to multiple viruses, including Rice stripe virus [169] and Rice black-streaked dwarf virus [170]. This approach demonstrates the potential for engineering broad-spectrum resistance in various crops.

2. Interference with Virus Replication: CRISPR-Cas (Clustered Regularly Interspaced Short Palindromic Repeats and CRISPR-associated proteins) technology has shown great promise in interfering with the replication of plant viruses. CRISPR-Cas allows for precise and targeted modifications of the genomic DNA of organisms, including plants, providing

a potential tool for developing resistance against viral infections. CRISPR-based approaches can be designed to interfere with key steps in the virus replication process. For example, targeting viral genes or regulatory elements can disrupt the virus's ability to replicate within the plant. One notable example of CRISPR-Cas technology application is to interfere with plant virus replication and the development of virus-resistant crops to the devastating geminiviruses, which infect a wide range of crops including tomatoes, cassava, and cotton [171]. scientists used CRISPR-Cas9 to engineer resistance in cassava plants by targeting the AC2/AC3 component of Cassava mosaic virus [172]. The edited plants showed reduced symptoms and viral DNA accumulation. Similarly, enhanced resistance against turnip mosaic virus (TuMV) infection in Arabidopsis thaliana was obtained by using CRISPR-Cas9 mediated interference with plant virus replication by designing guide RNAs against the viral genome [173]. Furthermore, researchers used CRISPR-Cas9 to target the bean yellow dwarf virus, beet severe curly top virus, tomato yellow leaf curl virus, cotton leaf curl Multan virus, wheat dwarf virus, chili leaf curl virus, etc. in common *Nicotiana benthamiana* plants [174]. By designing guide RNAs specific to the viral genome, they achieved targeted mutagenesis and observed a reduction in virus accumulation. These examples illustrate the broader applications of CRISPR technology in conferring resistance to plant diseases.

3. Enhancing Plant Immune Responses: CRISPR can be used to enhance the plant's natural immune responses against viruses. This may involve modifying genes that regulate the production of antiviral proteins or other defense mechanisms. CRISPR can also be employed to enhance the expression of genes associated with the plant's defense mechanisms. By increasing the production of proteins or signaling molecules involved in plant immunity, the plant's ability to resist infections can be bolstered [175]. Furthermore, PRRs play a crucial role in recognizing pathogen-associated molecular patterns (PAMPs) and triggering immune responses. CRISPR editing can be used to modify PRRs to enhance their sensitivity or specificity, making plants more responsive to potential threats.

In this way, CRISPR allows for the development of both broad-spectrum and specific resistance. Broad-spectrum resistance targets common features shared by multiple viruses, while specific resistance is tailored to a particular virus. CRISPR-based plant virus resistance holds great promise for improving crop yields and global food security by providing an efficient and targeted method for enhancing plant defense mechanisms against viral infections. Ongoing research in this field continues to refine and expand the applications of CRISPR technology in agriculture. The development and deployment of CRISPR-edited plants are subject to regulatory frameworks that vary by country. Ensuring safety and adherence to regulations is crucial before widespread adoption. Efforts are made to minimize off-target effects during CRISPR editing to ensure that unintended modifications do not occur in the plant genome. It's important to note that while these examples demonstrate the potential of CRISPR-Cas technology in developing virus-resistant crops, challenges remain.

## 4. Practical Application

### 4.1. Production of dsRNA for Plant Virus Management

In the field of molecular biology, nucleic acid extraction is a widely used technique to isolate and purify DNA or RNA molecules from biological samples. This process is essential for various downstream applications, including the generation of double-stranded RNA (dsRNA). dsRNAs are powerful tools for gene silencing and have extensive applications in plant research, particularly for studying gene function and plant–virus interactions [176]. To generate dsRNA, the first step is to obtain a DNA template that encodes the RNA sequence of interest. One approach to generate dsRNA is by using *Escherichia coli* HT115. In this method, RNA sequences are cloned into a plasmid vector, which is then transformed into *E. coli* HT115. The bacteria are induced to express the RNA sequences, which can be subsequently purified and converted into dsRNA [177,178]. Another approach to producing dsRNA is by utilizing plant-based systems, such as agro-inoculation. In this method, bacterial vectors carrying the desired RNA sequences are introduced into plant cells. The RNA sequence is cloned into a binary plasmid vector, which is then trans-

formed into *Agrobacterium tumefaciens*, a bacterium that naturally infects plants [179]. The transformed *Agrobacterium* is used to infect plant tissue, allowing for the expression of the RNA sequences within the plant. Different inoculation methods, such as leaf infiltration, stem injection, and root immersion, can be employed for agro-inoculation [180,181]. When generating dsRNA through agro-inoculation, careful consideration should be given to the choice of plant material, inoculum, and data analysis method. Different plant species exhibit varying susceptibility to agro-inoculation, and the efficiency of the transformation process may vary depending on the source of the inoculum [182]. Once dsRNA has been generated using either of these approaches, it is crucial to characterize its sequence and activity. Small RNA sequencing and analysis techniques enable the identification and quantification of small RNAs, including viral small interfering RNAs (vsiRNAs) [183]. Real-time quantitative PCR (qPCR) can be employed to quantitate dsRNA and vsiRNAs, using fluorescent probes to measure the amount of DNA or RNA in a sample [184]. Quantifying vsiRNA is particularly important for studying plant–virus interactions, as it allows researchers to evaluate the effectiveness of gene silencing and identify potential targets for antiviral strategies. By combining different techniques and approaches, researchers can gain a deeper understanding of the molecular mechanisms underlying plant-virus interactions and develop new tools for crop protection and genetic engineering [185].

### 4.2. Topical Application of Double-Stranded RNA to Protect Plants from Viral Infection

Exogenously applying RNA interference (RNAi)—inducing double-stranded RNA (dsRNA) shows promise for controlling infections caused by plant viruses. This method involves using dsRNAs that specifically target viral RNA sequences in plants. Furthermore, when aphids consume these dsRNAs, they trigger RNAi within the insects, leading to the degradation of viral RNA and ultimately inhibiting viral transmission [186]. When aphids feed on plants, they secrete saliva, which contains enzymes and compounds that facilitate feeding and viral transmission [187]. The ingested dsRNA molecules are believed to interfere with the expression of genes involved in viral replication and transmission within aphids. This degradation of viral RNA reduces the viral titer in aphids, making successful transmission to other plants less likely. Research has demonstrated that exogenous application of dsRNA effectively inhibits the aphid-mediated infection of plant viruses, including cucumber mosaic virus, potato virus Y, and bean yellow dwarf virus [188]. This method has also proven effective in controlling aphid populations by reducing feeding and egg production, leading to improved plant health and higher yields.

One challenge in utilizing RNAi-based approaches to control plant viral transmission is ensuring the specificity of the dsRNA molecules used. The dsRNA molecule must be carefully designed to target only viral RNA and avoid unintended effects on the plant's own RNA [187]. Overall, exogenous application of RNAi-inducing dsRNA molecules holds promise for managing the aphid-mediated transmission of plant viruses. This method is relatively simple, cost-effective, and could be a valuable tool in comprehensive pest management strategies aimed at protecting crops from viral infections [189].

This can be exemplified in aphid-mediated transmission and infection of bean common mosaic virus (BCMV). The dissected leaf assay is commonly employed to explore the role of dsRNA in the transmission of and infections caused by plant viruses [150]. In this assay, leaves isolated from infected plants are exposed to aphids that have fed on BCMV-infected plants [190]. The assay can be conducted using various aphid species, including the commonly used *Myzus persicae*. Aphids are allowed to feed on BCMV-infected plants to acquire the virus. Subsequently, they are transferred to the detached leaves of healthy plants, which are kept in a moist environment to facilitate aphid feeding. The leaves are then observed over several days for symptoms of BCMV infection, such as mosaic patterns, necrosis, and chlorosis [191].

By acquiring the virus from infected plants, aphids infect the detached leaves during feeding. The virus then replicates within the leaves, resulting in characteristic symptoms. The severity of symptoms serves as an indicator of the efficiency of virus transmission by

aphids [192]. Single-leaf assays are valuable for studying the mechanisms of viral infection by aphids and testing the efficacy of different control strategies. This method is relatively straightforward, requires minimal equipment, and can be performed on a wide range of plant and aphid species [193]. However, it should be noted that this method does not fully replicate the conditions of viral infection in natural environments, and the results can be influenced by factors such as plant age, health, temperature, humidity, environment, and genetic variability. Additionally, there are limitations related to the virus and aphid populations [194] (Table 4).

**Table 4.** Topical application of dsRNA on host plants for plant virus management.

| Plant Viral Disease | Cure with dsRNA Technology | Host Plant | References |
|---|---|---|---|
| Potato virus Y | Introduction of dsRNA targeting the viral coat protein gene | Potato (*Solanum tuberosum*), | [195] |
| Tomato spotted wilt virus | Expression of dsRNA targeting the viral nucleocapsid gene | Tomato (*Solanum lycopersicum*) | [196] |
| Cucumber mosaic virus | Application of dsRNA targeting viral replicase genes | Solanaceous | [197] |
| Tobacco mosaic virus | Expression of dsRNA targeting viral coat protein genes | Tobacco (*Nicotiana tabacum*) | [198] |

*4.3. RNAi to Control Insect Vectors*

Double-stranded RNA (dsRNA) is a genetic material with the ability to selectively silence genes in various organisms, including insects [199]. The use of dsRNA as a tool for pest management has been explored by scientists, but it has faced challenges such as limited efficacy and off-target effects. However, a recent breakthrough technique developed by researchers at the University of Maryland has shown promising results in controlling pests. The technique involves encapsulating dsRNA in nanoparticles that are ingested by insect predators, such as ladybugs and lacewings [159]. Once the dsRNA is released in the predator's gut, it silences genes crucial for the survival of targeted pests like the Colorado potato beetle or the diamondback moth. As a result, the pest population is reduced when the predator feeds on them. This technique specifically targets pests without harming non-target organisms and eliminates the need for broad-spectrum insecticides, which can be environmentally harmful [152]. Using dsRNA as a pest management tool has the potential to offer a more sustainable and eco-friendly alternative to traditional methods. However, further research is required to optimize dsRNA delivery and ensure its safety and efficacy in real-world applications.

In the field of agriculture, RNA interference (RNAi) has been utilized for pest and pathogen control through methods such as host-induced gene silencing (HIGS), spray-induced gene silencing (SIGS), and virus-induced gene silencing (VIGS). HIGS involves creating transgenic plants that express pest- or pathogen-specific dsRNAs [154]. The first commercially available RNAi product targeted against pests was a transgenic maize plant developed by Monsanto (now Bayer Crop Science), which expressed dsRNA targeting the snf7 gene of the western maize rootworm Diabrotica virgifera [200]. This RNAi construct also contained two Bacillus thuringiensis Cry proteins (Cry3Bb1 and Cry34/35Ab) to delay resistance development [200]. Marketed as Smart Stax Pro, the product was approved by the U.S. Environmental Protection Agency in 2017 and is expected to be commercially available by the end of the decade, marking a significant milestone in the application of RNAi technology in agriculture [201,202].

Virus-based expression vectors are another option to produce desired dsRNA within the host itself. The plant virus-derived expression systems [203–208] present a multitude of benefits, encompassing elevated expression levels, swift production, scalability, safety, and economical feasibility. Viral infection and replication directly generate dsRNA molecules in insect cells. Notably, Flock House virus (FHV) has been successfully engineered to

express Drosophila melanogaster-specific dsRNA [209]. Other techniques propose the use of various microorganisms, including bacteria, yeast, or fungi, engineered as vectors to induce gene silencing by continuously producing si/dsRNA within host cells [185,210,211]. A review covering the use of bacteria and viruses for dsRNA delivery is available [185]. The utilization of microbes or derived products for insect and disease management is discussed in subsequent sections, focusing on their potential, achievements, and concerns. Moreover, efforts are being made to employ non-transgenic, spray-based insecticide dsRNA (SIGS) to control pests and pathogens. SIGS can also be applied through root uptake and stem injection, allowing insects to acquire dsRNA through feeding mechanisms like sucking or chewing. An overview of this management method is provided in [185] (Figure 3).

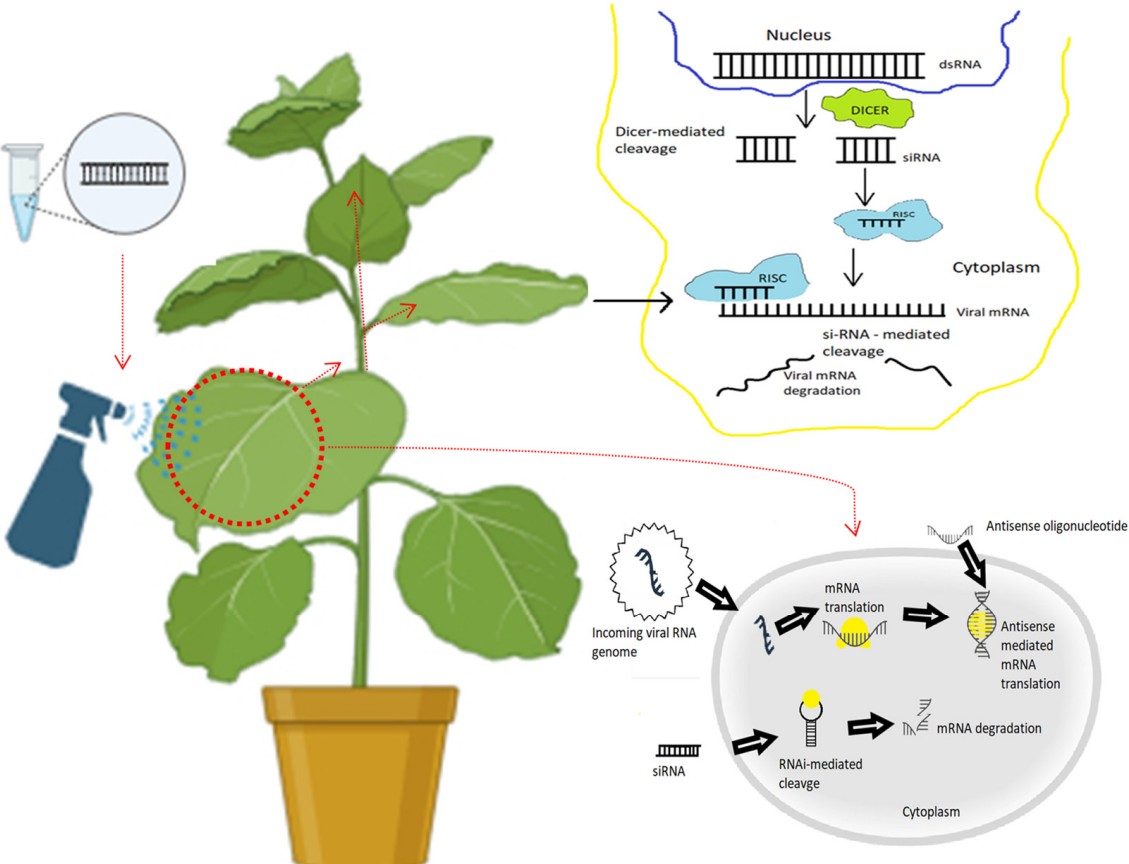

**Figure 3.** RNA interference (RNAi)-mediated plant defense against virus pathogens. In vitro synthesis of dsRNA followed by delivery into the plant leads to the management of plant viruses through RNAi in the delivered tissue as well as in the systemic tissue. The red arrow indicates the inoculation of dsRNA into plants using spray method.

## 5. Conclusions and Future Perspectives

In summary and looking to the future, the combination of RNA interference (RNAi) and double-stranded RNA (dsRNA) technologies offers a promising avenue for the sustainable and user-friendly control of plant viruses. The economic importance of Phyto-viruses, characterized by significant crop losses, emphasizes the urgency of strengthening our arsenal against these biotic stressors. Although considerable progress has been made in using exogenous RNA and dsRNA technologies to precisely target viral genes, the journey is far from over.

In the future, a promising roadmap will be revealed. Future research should focus on fine-tuning and perfecting the delivery of exogenous RNA and dsRNA to plant cells, overcoming barriers such as cell walls and membranes while ensuring the stability of the environment. To ensure effectiveness, the choice of RNA molecule, including its sequence

and length, must be carefully considered. Additionally, it is essential to decide the proper exogenous RNA or dsRNA dosage to maximize uptake by target plant cells. Equally important is the choice of target genes for silencing; finding conserved regions in viral genomes important for replication or transmission will improve the efficiency of gene silencing and virus control.

To counter potential challenges, vigilance in monitoring viral populations is essential to detect any emergence of drug resistance or changes in viral diversity. This information will guide the development of strategic countermeasures, which are likely to involve the deployment of multiple targets or integrated approaches that often combine RNAi technology with screening or antiviral agents. In this era of agricultural transformation, exogenous RNA and dsRNA technology promises to reduce our dependence on chemical pesticides, minimize crop damage, and improve the overall sustainability of agriculture. As we look to the future, continued research and innovation in this field will drive the creation of effective and practical solutions to manage viral diseases in agriculture, thereby ensuring food safety and whole food and environmental health.

**Author Contributions:** B.M.D., S.G., P.B., A.C., S.S.T. and A.A.K.J.; writing—original draft preparation, B.M.D., S.G., P.B., A.C. and A.A.K.J.; figures and tables, S.G., A.C. and A.A.K.J., review and editing, B.M.D., A.C., K.V.D., M.C., S.M., S.S.T. and A.A.K.J.; supervision, critical comments and suggestions, and manuscript revision, A.C., S.S.T., S.M. and A.A.K.J.; funding acquisition, S.M. and A.A.K.J. All authors have read and agreed to the published version of the manuscript.

**Funding:** This research received no external funding.

**Institutional Review Board Statement:** Not applicable.

**Informed Consent Statement:** Not applicable.

**Data Availability Statement:** Not applicable.

**Conflicts of Interest:** The authors declare no conflicts of interest.

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
