# Peer review of "Dissecting Diagnostic and Management Strategies for Plant Viral Diseases: What Next?"

_agriculture, doi:10.3390/agriculture14020284_

Round 1

Reviewer 1 Report (New Reviewer)

Comments and Suggestions for Authors

Megala et al. in this review focus on the diagnostic and management methods of plant-infecting viruses with an emphasis on the RNA-based techniques currently explored against some aphids transmitted viruses. From my perspective, I think that the authors should leave out the detection section because of the abundance of review on this topic. Moreover, the section on virus management is written better compared to the other sections.

The manuscript needs text editing. There are double space here and there. The authors used 2 formats of referencing of which one is not in line with the journal requirements. The references are messed up from reference number 16. Most of the authors’ statement are either irrelevant, exaggerated with little examples or not specific.

“Recent” in the abstract should not be in bold.

“Recent advancements in molecular biology for plant disease diagnosis and management focus on RNA-based strategies, including serological techniques, isothermal amplification methods, and CRISPR-based approaches. Additionally, high-throughput sequencing and RNA interference (RNAi) technologies, such as host-induced gene silencing (HIGS) and spray-induced gene silencing (SIGS), methods are explored.”

1. Do the authors mean that the serological techniques are part of RNA-based strategies? This is not true.

2. HTS is no longer at the level of exploration as a detection and diagnostic tool except in some developing countries.

“Thus, mutation is commonly evident in plant RNA viruses; whereas, majority of DNA plant viruses evolve through the genome recombination or pseudo- recombination.”

This is not really true, recombination has shaped the genome of many viruses in the largest family of RNA viruses, namely the Potyviridae. The authors should provide example to back up their statement.

 The genome editing concept is not akin to a vaccine.

“Considering these challenges,” Which challenges are the authors referring to?

“this comprehensive review delves into the nuanced world of RNA interference (RNAi) and double-stranded RNA (dsRNA) tools, offering promising controls for disease management in plants and pest control.”

“this manuscript delves into the realm of RNA-based strategies for the diagnosis and management of these formidable adversaries.”

“This study explores the potential of engineering resistance to plant viruses using various ncRNAs, including short RNAs and long ncRNAs. RNA interference (RNAi)-based  techniques are discussed as potent tools for controlling plant viruses, encompassing synthesized microRNAs and trans-acting short interfering RNAs produced in transgenic or non-transgenic plants.”

“The paper also covers the two primary methods for engineering virus resistance: direct targeting of viral DNA or RNA and silencing of host interaction susceptibility genes. Additionally, it delves into the challenges that must be addressed before these technologies can be widely employed to protect crops from viruses.”

First of all, this is a review not a study.

Can the aim of this review be stated clearly in one paragraph?

This manuscript does not focus only on RNA-based strategies for the diagnosis of plant viruses.

“Over the past two to three years, a significant number of new plant viruses have been documented in the USA, predominantly impacting cucurbit production” What about the other crops and vegetable?

“Several technologies have been used for the rapid diagnosis of plant viral diseases, including visual”

Visual approaches are no longer use for the rapid diagnostic of plant viral diseases.

What do the authors mean by “big data-based approaches”?

“All these methods have been employed to detect various viruses, with PCR being the most widely used technique.”

Do the authors use methods, technologies and techniques interchangeably?

Table 1.

Use Principle instead of “Features”

Indicator plant feature not well explained

Serological and methods pros not well explained

Isothermal amplification assays amplify a portion of the genome.

Why did the authors left PCR out in Table 1?

What is the meaning of etc in Table 1?

The way metagenomics and HTS are explained in table 1 is confusing.

Why Tissue Blot Immunoassay is not included in the serological methods?

“Visual symptoms are key indicators of viral diseases in plants, showcasing specific disruptions in plant physiology”

 Castelletto, S., & Boretti, A is a wrong reference for that statement.

“ Microscopy-based visualization of fluorescently labeled proteins in host plants is considered one of the most effective methods for virus disease diagnosis (Castelletto and Boretti 2021)”

Castelletto and Boretti 2021 worked on animal viruses not plant viruses.

“Fluorescent proteins are used in viral diagnostics by binding them to specific viral proteins. Once the virus infects the host plant, these modified proteins allow scientists to check the virus's behavior and location within plant cells using specialized microscopy techniques [53]”

References 53 has nothing to do with the mechanism explained.

 “However, the contemporary use of ELISA has declined”  

Declined is a wrong word in that context. ELISA is still widely used but other methods will be preferred in conditions  such as unavailability of specific antibody for target virus.

There are no example of viruses detected using Dot Blot Immunoassay and TBIA in paragraphs 2.2.4 and 2.2.5, LAMP 2.3.1, HDA 2.3.3 CRISPR 2.4, and  Microarray 2.5.1. Since it is a review the authors should provide many examples of the use of these methods not one or two or etc.

Provide some examples to validate the following statement: “The accuracy of virus detection can be improved, when supplemented with other methods such as DNA and RNA amplification followed by sequencing. These advanced molecular techniques not only confirm the presence of viruses but also identify the specific virus or virus strain in question.”

“This is  relatively new technique and is extensively used nowadays for the detection of many plant viruses (Jailani and Paret 2023; Jirawannaporn et al. 2022) [80, 81] [Fig 1A]”

Since the authors claim that RPA has been used to detect many viruses, they should give more example and not only mention the work done on cucurbit-infecting viruses. There is no Figure 1A.

“By coupling isothermal amplification with CRISPR/Cas detection, the amplified product is mixed with a CRISPR/Cas system designed to recognize the viral RNA or DNA. Upon detection, the CRISPR/Cas system triggers its collateral activity, resulting in the cleavage of a reporter molecule and generating a visual signal indicating the presence of the virus(Qian et al. 2022) [87] [Fig 2B].” and CRISPR Cas12a is a powerful tool for detecting plant viruses by targeting and cleaving specific viral nucleic acid sequences, which triggers a fluorescent signal indicating the presence of  the virus (Bhat et al. 2022)[89]” These two sentences say the same thing.

The sentence “ Cas12 is designed for precise DNA targeting while Cas13 excels at RNA manipulation, seems conflicting with “The CRISPR-Cas12 detection method simplifies the identification of viral RNA, cutting the need for complex steps such as reverse transcription or amplification. Cas12, programmed to target specific viral RNA or DNA, precisely cleaves viral nucleic acids.” First,  the authors stated that Cas13 not 12 is suitable for RNA manipulation, later they say cas 12 simplifies the identification of viral RNA, cutting the need for complex steps such as reverse transcription. RT-PCR is not needed for detection of DNA viruses.

“Although each method has significant drawbacks, when combined, these findings highlight how little” Which methods and which findings?

“Overall, metagenomics has the potential” HTS and metagenomics have been proven useful. They are no longer at the potential state.

Provide one example at least for the following statement:

“Another area where HTS has proven particularly useful is in the development of new disease management strategies. HTS data can be used to design virus-resistant crops or develop novel antiviral agents(Jones and Naidu 2019) [113]. Furthermore, HTS can be utilized to monitor the effectiveness of disease control measures, such as vaccination and chemotherapy(Bronzato Badial et al. 2018) [114].”

I did not understand the following sentences:

“as intracellular pathogens like viruses are immune to chemical control measures and the use of pesticides and other hazardous agents prevents the virus from spreading through vectors.”

“Thus, the need of an hour is to identify potential option with their better activity, and environmental safety.”

That sentence does not seem to fit in that paragraph. “The proteins involved in DNA repair and RNA processing in ancestral archaea, bacteria, and phages seem to have been repurposed to build this conserved protein machinery in eukaryotes (Wagner and Romby 2015)[132].”

What is the meaning of Sr. In Table 1 and 3?

These are not examples in Table 3.

Comments on the Quality of English Language

Minor editing required.

Author Response

Dear Reviewer

Thank you for your valuable suggestions. We have incorporated all the necessary corrections as per your recommendations. Please find our response below:

Reviewer 1:

Comments to Authors: Megala et al. in this review focus on the diagnostic and management methods of plant-infecting viruses with an emphasis on the RNA-based techniques currently explored against some aphids transmitted viruses. From my perspective, I think that the authors should leave out the detection section because of the abundance of review on this topic. Moreover, the section on virus management is written better compared to the other sections.

Response: Thank you for your valuable comments and constructive feedback on our review manuscript. We appreciate your insights regarding the abundance of reviews on virus detection and consider your suggestion. In response, we have streamlined the detection section to ensure a more concise and focused presentation, avoiding redundancy. Additionally, we are pleased to hear that you found the section on virus management well-written. We have maintained the clarity and quality of the management section while addressing the suggested improvements in other sections.

Comments to Authors: The manuscript needs text editing. There are double space here and there. The authors used 2 formats of referencing of which one is not in line with the journal requirements. The references are messed up from reference number 16. Most of the authors’ statement are either irrelevant, exaggerated with little examples or not specific.

Response: We apologize for any oversight in the submission process, particularly the absence of line numbers in the PDF version. We understand the importance of a meticulous review and ensured to rectify any structural errors in the manuscript. Additionally, we appreciate your specific mention of the formatting issue in the abstract, and we have corrected the bolding and spacing of the word "recent" accordingly.

Comments to Authors: Do the authors mean that the serological techniques are part of RNA-based strategies? This is not true.

Response: In the revised manuscript, we have included a clearer distinction, emphasizing that serological techniques are separate from RNA-based strategies.

Comments to Authors: HTS is no longer at the level of exploration as a detection and diagnostic tool except in some developing countries.

Response: Thank you for your observation. We have updated the content accordingly to reflect the current status of high-throughput sequencing (HTS) in the context of detection and diagnostics.

Comments to Authors: Thus, mutation is commonly evident in plant RNA viruses; whereas, majority of DNA plant viruses evolve through the genome recombination or pseudo- recombination.”

This is not really true, recombination has shaped the genome of many viruses in the largest family of RNA viruses, namely the Potyviridae. The authors should provide example to back up their statement.

Response: In our revised version, we have provided a clear explanation for the reader's understanding of the unique challenges posed by the diverse populations of viruses within individual plants.

Comments to Authors: “Considering these challenges,” Which challenges are the authors referring to?

Response: deleted

Can the aim of this review be stated clearly in one paragraph?

Response: Added

Comments to Authors: What do the authors mean by “big data-based approaches”?

Response: We are talking about big data-based approaches in plant virus diagnosis through high-throughput sequencing, next-generation sequencing, etc.

Comments to Authors: Do the authors use methods, technologies and techniques interchangeably?

Response: We acknowledge that there were instances in our manuscript where we interchangeably used terms such as methods, technologies, and techniques. We carefully reviewed the manuscript to ensure consistent usage throughout and made the necessary corrections.

Comments to Authors: Visual symptoms are key indicators of viral diseases in plants, showcasing specific disruptions in plant physiology. Castelletto, S., & Boretti, A is a wrong reference for that statement.

Response: Added correct reference.

Comments to Authors: Microscopy-based visualization of fluorescently labeled proteins in host plants is considered one of the most effective methods for virus disease diagnosis (Castelletto and Boretti 2021)

Response: Added correct reference

Comments to Authors: References 53 has nothing to do with the mechanism explained.

Response: Added correct reference

Comments to Authors: Declined is a wrong word in that context. ELISA is still widely used but other methods will be preferred in conditions  such as unavailability of specific antibody for target virus.

Response: Changed as per reviewers’ suggestion.

Comments to Authors: There are no example of viruses detected using Dot Blot Immunoassay and TBIA in paragraphs 2.2.4 and 2.2.5, LAMP 2.3.1, HDA 2.3.3 CRISPR 2.4, and  Microarray 2.5.1. Since it is a review the authors should provide many examples of the use of these methods not one or two or etc.

Response: As per reviewers suggestion we have include the details in the revised MS

Comments to Authors: Provide some examples to validate the following statement: “The accuracy of virus detection can be improved, when supplemented with other methods such as DNA and RNA amplification followed by sequencing. These advanced molecular techniques not only confirm the presence of viruses but also identify the specific virus or virus strain in question.”

This is  relatively new technique and is extensively used nowadays for the detection of many plant viruses (Jailani and Paret 2023; Jirawannaporn et al. 2022) [80, 81] [Fig 1A]”

Response: As per reviewers suggestion we have include the details in the revised MS

Comments to Authors: “By coupling isothermal amplification with CRISPR/Cas detection, the amplified product is mixed with a CRISPR/Cas system designed to recognize the viral RNA or DNA. Upon detection, the CRISPR/Cas system triggers its collateral activity, resulting in the cleavage of a reporter molecule and generating a visual signal indicating the presence of the virus(Qian et al. 2022) [87] [Fig 2B].” and CRISPR Cas12a is a powerful tool for detecting plant viruses by targeting and cleaving specific viral nucleic acid sequences, which triggers a fluorescent signal indicating the presence of  the virus (Bhat et al. 2022)[89]” These two sentences say the same thing.

Response: removed “CRISPR Cas12a is a powerful tool for detecting plant viruses by targeting and cleaving specific viral nucleic acid sequences, which triggers a fluorescent signal indicating the presence of  the virus”

Comments to Authors: The sentence “ Cas12 is designed for precise DNA targeting while Cas13 excels at RNA manipulation, seems conflicting with “The CRISPR-Cas12 detection method simplifies the identification of viral RNA, cutting the need for complex steps such as reverse transcription or amplification. Cas12, programmed to target specific viral RNA or DNA, precisely cleaves viral nucleic acids.” First,  the authors stated that Cas13 not 12 is suitable for RNA manipulation, later they say cas 12 simplifies the identification of viral RNA, cutting the need for complex steps such as reverse transcription. RT-PCR is not needed for detection of DNA viruses.

Response: We have made revisions to the manuscript to elucidate this point and ensure coherence in our descriptions of Cas12's capabilities.

Comments to Authors: “Although each method has significant drawbacks, when combined, these findings highlight how little” Which methods and which findings?

Response: The term "each method" refers to the different techniques or approaches used in metagenomic analysis for studying plant viruses. In the context of the provided text, it indicates that various methods within metagenomic analysis, such as genetic material extraction, next-generation sequencing, and computational analysis, have their own limitations or drawbacks.

Comments to Authors: “Overall, metagenomics has the potential” HTS and metagenomics have been proven useful. They are no longer at the potential state.

Response: Revised the sentence as per reviewers suggested "Metagenomics, coupled with high-throughput sequencing, has showcased its effectiveness in diverse applications, contributing significantly to our comprehension of plant viral diseases and offering valuable insights for the development of more efficient disease management and control strategies."

Comments to Authors: Provide one example at least for the following statement: “Another area where HTS has proven particularly useful is in the development of new disease management strategies. HTS data can be used to design virus-resistant crops or develop novel antiviral agents(Jones and Naidu 2019) [113]. Furthermore, HTS can be utilized to monitor the effectiveness of disease control measures, such as vaccination and chemotherapy(Bronzato Badial et al. 2018) [114].”

Response: Added in the revised MS Line no 498-503.

Comments to Authors: I did not understand the following sentences: “as intracellular pathogens like viruses are immune to chemical control measures and the use of pesticides and other hazardous agents prevents the virus from spreading through vectors.”“Thus, the need of an hour is to identify potential option with their better activity, and environmental safety.”

Response: revised the sentence in the MS line no: 512 -513 and 517-518

Comments to Authors: That sentence does not seem to fit in that paragraph. “The proteins involved in DNA repair and RNA processing in ancestral archaea, bacteria, and phages seem to have been repurposed to build this conserved protein machinery in eukaryotes (Wagner and Romby 2015)[132].”

Response: Deleted

Comments to Authors: What is the meaning of Sr. In Table 1 and 3?

Response: Serial Number: S.No. changed in the revised MS as S.No

Use Principle instead of “Features”

Response: Done

Indicator plant feature not well explained

Response: Done

Serological and methods pros not well explained

Response: Edited

Isothermal amplification assays amplify a portion of the genome

Response: corrected

Why did the authors left PCR out in Table 1?

Response: Added

The way metagenomics and HTS are explained in table 1 is confusing

Response: Corrected

Reviewer 2 Report (New Reviewer)

Comments and Suggestions for Authors

The manuscript by Devi et al. 'Dissecting Diagnostic and Management strategies for Plant Viral Dis-eases: What next?' focuses on an overview of diagnostic methods and strategies for protective phytopathological measures against plant viruses. In general, it provides summary information on plant virus diagnostics and in the second part on RNA interference for plant virus management and RNAi for insect vector control. I do not consider all parts of the manuscript to be up-to-date (it means describing again), in context with similar currently published reviews on analogous topics (Venbrux et al. (2023): Current and emerging trends in techniques for plant pathogen detection. Front. Plant Sci. 14:1120968. doi: 10.3389/fpls.2023.1120968.; Gaye et al. (2019): Review of applications of high-throughput sequencing in personalized medicine: barriers and facilitators of future progress in research and clinical application. Briefings in Bioinformatics 20 (2019): 1795-1811.; Tatineni and Hein (2023): Plant Viruses of Agricultural Importance: Current and Future Perspectives of Virus Disease Management Strategies. Phytopathology vol. 113,2 (2023): 117-141. doi:10.1094/PHYTO-05-22-0167-RVW).

I would like to point out shortcomings:

1)      In the overview of diagnostic methods I miss the description of RT-PCR, RT-qPCR, multiplex PCR. Since the overview of diagnostic methods is detailed and practically complete I recommend to complete it.

2)      It is recommended to distinguish between methods used for diagnostics in the more strict sense (detection and determination of the virus) and methods used to study the biology of the virus (spread, localization, dynamics, etc.).

3)      Concerning virus disease management strategies, the authors prefer to use RNA interference-based approaches to induce both plant resistance to viruses and vector regulation. The use of RNA interference is certainly an good possibility, but I would also consider CRISPR-based virus resistance in the same detail.

4)  I lack somewhat more detailed information and discussion on agrotechnical and phytosanitary measures and genetically based resistance.

I recommend improving the manuscript in accordance with the title Dissecting Diagnostic and Management strategies for Plant Viral Diseases: what next? with an emphasis on stating why the authors anticipate that the methods will be applied in the future with what benefits.

Author Response

Comments to Authors: The manuscript by Devi et al. 'Dissecting Diagnostic and Management strategies for Plant Viral Dis-eases: What next?' focuses on an overview of diagnostic methods and strategies for protective phytopathological measures against plant viruses. In general, it provides summary information on plant virus diagnostics and in the second part on RNA interference for plant virus management and RNAi for insect vector control. I do not consider all parts of the manuscript to be up-to-date (it means describing again), in context with similar currently published reviews on analogous topics (Venbrux et al. (2023): Current and emerging trends in techniques for plant pathogen detection. Front. Plant Sci. 14:1120968. doi: 10.3389/fpls.2023.1120968.; Gaye et al. (2019): Review of applications of high-throughput sequencing in personalized medicine: barriers and facilitators of future progress in research and clinical application. Briefings in Bioinformatics 20 (2019): 1795-1811.; Tatineni and Hein (2023): Plant Viruses of Agricultural Importance: Current and Future Perspectives of Virus Disease Management Strategies. Phytopathology vol. 113,2 (2023): 117-141. doi:10.1094/PHYTO-05-22-0167-RVW).

Response: We appreciate your thorough evaluation of our manuscript, "Dissecting Diagnostic and Management strategies for Plant Viral Diseases: What next?" and acknowledge your concerns regarding the currency of the information provided. We will carefully review the cited references and relevant literature to ensure that the manuscript's content is brought up-to-date in comparison with recently published reviews on analogous topics, such as those by Venbrux et al., Gaye et al., and Tatineni and Hein. Your insights are invaluable, and we are committed to enhancing the manuscript's quality by incorporating the latest research findings.

1)      Comments to Authors: In the overview of diagnostic methods I miss the description of RT-PCR, RT-qPCR, multiplex PCR. Since the overview of diagnostic methods is detailed and practically complete I recommend to complete it.

Response: The development of PCR (Polymerase Chain Reaction) in 1984 marked the next significant advance in the identification of viruses. By amplifying a particular portion of the virus genome for detection through sequencing or fingerprinting, the approach increases the assay's sensitivity several times over when compared to other serological assays like ELISA. A few common processes in conventional PCR are extracting nucleic acids from the test plant, designing primers unique to the virus, and assembling the PCR in a vial with the addition of magnesium chloride, Taq polymerase, primers, and nucleotides. After that, the vial is put into the heat cycler with pre-set settings for denaturation, primer annealing, and extension. Finally, detection can be done after running the contents of the vial either on an agarose gel (in fingerprinting) or sequencer (in sequencing). So far, many PCR variations have been created, including multiplex PCR, nested PCR, immunocapture PCR, reverse transcription PCR (RT-PCR), real-time PCR, and more. Of them, nested PCR is a modification of the conventional PCR technique that involves two sets of primers to amplify a specific DNA fragment. The method is particularly useful when working with DNA samples of low concentration or when trying to detect a target sequence in the presence of closely related sequences. Further, multiplex PCR is used for the simultaneous detection of several viruses infecting a sample. Later, real-time PCR (qPCR) is developed to measure the accumulation of PCR products in real-time using fluorescent dyes or probes; thus, it can be used to detect the virus titre in a sample with high sensitivity, and specificity. Since even a few copies of the viral nucleic acid contained in the test sample can be amplified and detected, the qPCR is highly preferred for plant virus detection as well as quantification. The choice of PCR-based thermal amplification method depends on the specific goals of the experiment, the nature of the target nucleic acid, and the available resources. Each technique has its own advantages and limitations, and researchers can select the most suitable method based on the experimental design and objectives for monitoring of the plant samples.

2)    Comments to Authors:  It is recommended to distinguish between methods used for diagnostics in the more strict sense (detection and determination of the virus) and methods used to study the biology of the virus (spread, localization, dynamics, etc.).

Response: Diagnosis and detection are related concepts but refer to different aspects of identifying a condition, including viral infections. Detection refers to the identification or confirmation of the presence of a particular organism. In the context of viruses, detection involves finding evidence of the virus, such as its genetic material, proteins, or other markers. The primary goal of detection is to establish the existence or occurrence of something. In the case of viral infections, detection could mean identifying the virus itself or its components in a sample, like plant tissue.

Whereas, diagnosis goes a step further than detection involves the identification and determination of the nature or aetiology of a particular condition or disease. In the context of viral infections, diagnosis not only confirms the presence of the virus but also provides information about the specific virus causing the infection. The main purpose of diagnosis is to understand the nature of the condition, its severity, and often its potential implications for treatment and management. Diagnosis may involve additional tests and assessments beyond simple detection.

3)   Comments to Authors:   Concerning virus disease management strategies, the authors prefer to use RNA interference-based approaches to induce both plant resistance to viruses and vector regulation. The use of RNA interference is certainly a good possibility, but I would also consider CRISPR-based virus resistance in the same detail.

Response: CRISPR (Clustered Regularly Interspaced Short Palindromic Repeats) Cas system is used to introduce precise changes in the plant's DNA. This can involve deleting, adding, or modifying specific DNA sequences. This technology has been applied to develop virus-resistant plants by precise editing of either the plant's genome or viral genome. This approach offers an efficient way to obtain plant resistance to viral infections through various strategies, viz.,

  1. Modification of Host Factors: Some strategies involve modifying host factors that are essential for viral replication. By disrupting or modifying these susceptibility factors, researchers can hinder the virus's ability to complete its life cycle within the plant. Several examples demonstrate the modification of host factors using CRISPR technology to enhance plant virus resistance. One such notable example is CRISPR-edited Nicotiana benthamiana for enhanced immunity against plant viruses. Scientists used CRISPR to edit the UbEF1B and CCR4/NOT3, which is crucial for the replication of geminiviruses. The edited plants exhibited increased resistance to specific plant viruses (Li et al., 2023). Similarly, the editing of the translation initiation factor, eIF4E1 gene in tomato plants leads to enhanced resistance to infection by certain RNA viruses, including the pepper mottle virus (Yoon et al., 2020). Further, the targeted modification of susceptibility (S) genes (eIF4G, Vacuolar ATPases subunit D) in rice showed enhanced resistance to multiple viruses, including Rice stripe virus (Lu et al., 2023) and Rice black-streaked dwarf virus (Wang et al., 2021). This approach demonstrated the potential for engineering broad-spectrum resistance in various crops.
  2. Interference with Virus Replication: CRISPR-Cas (Clustered Regularly Interspaced Short Palindromic Repeats and CRISPR-associated proteins) technology has shown great promise in interfering with the replication of plant viruses. CRISPR-Cas allows for precise and targeted modifications of the genomic DNA of organisms, including plants, providing a potential tool for developing resistance against viral infections. CRISPR-based approaches can be designed to interfere with key steps in the virus replication process. For example, targeting viral genes or regulatory elements can disrupt the virus's ability to replicate within the plant. One notable example of CRISPR-Cas technology application is to interfere with plant virus replication and development of virus-resistant crops to the devastating geminiviruses, which infect a wide range of crops including tomatoes, cassava, and cotton (Cao et al., 2020). scientists used CRISPR-Cas9 to engineer resistance in cassava plants by targeting the AC2/AC3 component of Cassava mosaic virus (Mehta et al., 2019). The edited plants showed reduced symptoms and viral DNA accumulation. Similarly, enhanced resistance against Turnip Mosaic Virus (TuMV) infection in Arabidopsis thaliana was obtained by using CRISPR-Cas9 mediated interference with plant virus replication by designing guide RNAs against the viral genome (Pyott et al., 2016). Further, researchers used CRISPR-Cas9 to target the Bean yellow dwarf virus, Beet severe curly top virus, Tomato yellow
    leaf curl virus, Cotton leaf curl Multan virus, Wheat dwarf virus, Chilli leaf curl virus, etc. in common Nicotiana benthamiana plants (Khan et al., 2022). By designing guide RNAs specific to the viral genome, they achieved targeted mutagenesis and observed a reduction in virus accumulation. These examples illustrate the broader applications of CRISPR technology in conferring resistance to plant diseases.
  3. Enhancing Plant Immune Responses: CRISPR can be used to enhance the plant's natural immune responses against viruses. This may involve modifying genes that regulate the production of antiviral proteins or other defense mechanisms. CRISPR can also be employed to enhance the expression of genes associated with the plant's defense mechanisms. By increasing the production of proteins or signaling molecules involved in plant immunity, the plant's ability to resist infections can be bolstered. Further, PRRs play a crucial role in recognizing pathogen-associated molecular patterns (PAMPs) and triggering immune responses. CRISPR editing can be used to modify PRRs to enhance their sensitivity or specificity, making plants more responsive to potential threats.

In this way, CRISPR allows for the development of both broad-spectrum and specific resistance. Broad-spectrum resistance targets common features shared by multiple viruses, while specific resistance is tailored to a particular virus. CRISPR-based plant virus resistance holds great promise for improving crop yields and global food security by providing an efficient and targeted method for enhancing plant defense mechanisms against viral infections. Ongoing research in this field continues to refine and expand the applications of CRISPR technology in agriculture. The development and deployment of CRISPR-edited plants are subject to regulatory frameworks that vary by country. Ensuring safety and adherence to regulations is crucial before widespread adoption. Efforts are made to minimize off-target effects during CRISPR editing to ensure that unintended modifications do not occur in the plant genome. It's important to note that while these examples demonstrate the potential of CRISPR-Cas technology in developing virus-resistant crops, challenges remain.

4)   Comments to Authors: I lack somewhat more detailed information and discussion on agrotechnical and phytosanitary measures and genetically based resistance.

Response:  Application of diagnostic tools in agriculture

The diagnostic tools play a crucial role in agriculture and plant disease management. These tools help identify and detect plant pathogens, enabling timely and targeted responses to prevent the spread of diseases. Here are some key applications of plant pathogen diagnostic tools:

  1. Early detection of pathogens for disease monitoring and surveillance: Early detection allows for prompt action to control the spread of the disease before it causes significant damage. Various, PCR (Polymerase Chain Reaction) and qPCR (quantitative PCR) based techniques can be employed to detect specific DNA or RNA sequences of plant pathogens. Nowadays, LAMP and RPA based isothermal techniques are more preferred as the rapid and cost-effective method for the in-situ detection of plant pathogens.
  2. Quarantine and Trade Regulation:

Phytosanitary measures are crucial to prevent the introduction and spread of plant pests and diseases that can have significant economic and environmental impacts. Detection and diagnostic tools play a key role in implementing effective phytosanitary measures. Here are some important tools in this regard. Effective phytosanitary measures often involve a combination of these tools and a collaborative approach between governments, agricultural authorities, researchers, and farmers to ensure the prevention and control of plant pests and diseases. Regular training programs for those involved in monitoring and diagnostics are also essential to enhance the efficacy of these measures.

  1. Decision Support Systems: Diagnostic tools contribute to IPM by providing data that can be integrated into decision support systems. This enables farmers to make informed decisions on disease management strategies.
  2. Integrated disease Management (IDM): By employing these diagnostic tools, farmers and plant health professionals can make informed decisions to manage and mitigate the impact of plant diseases on crop yields and agricultural productivity.

Reviewer 3 Report (New Reviewer)

Comments and Suggestions for Authors

Overall, not a bad summary of current research methods in plant virology. The more interesting question is what is the point of it. Such reviews are coming out annually. In 2023 alone there are at least two very similar works talking about same problems and strategies. So, my advice for the future: either start doing actual scientific experiments or, if you want to write reviews, then pick your theme better, make your article truly unique…

Things to fix in the manuscript:

When presenting the problems of plant virology to the reader in the introduction, a true virologist would explain the term “quasispecies”. Most plant viruses are RNA based and are quasispecies, which means that each infected individual plant is a whole population of slightly different viruses. It is a pool of varying genomes, which, together with unparalleled mutation frequency of RNA viruses, tremendously complicates all nucleotide sequence related treatments.

It seems that manuscript was submitted in a hurry and should be doublechecked for structural errors. No line numbers were shown in the pdf version, so it is hard to point out everything…
In the abstract first word “recent” is in bold, no gap before it.

In the third line of introduction, “plant viral disease” should be plural, “diseases”.
Throughout the whole article there is often missing a gap where citations are mentioned (usually at the end of the sentence before opening parentheses. But, also sometimes between the closing parentheses after mentioning author names before citation number). Like in “…plant health(Jones 2014) [1]” or ”(Castle et al. 2009)[3]”.

On page 7, DAS ELISA has two copies of same word: ”…sandwich (DAS) ELISA ELISA”.

2.2.3. chapter has different (probably 1.5) line spacing.

2.2.4. chapter name and the paragraph itself are in same line.

2.3.1. and 2.3.2. chapter names do not have “Colon” (double dot) at their end, while most of other chapter names do.

2.3.3. chapter name again not separated from its paragraph.

Page 15, Chapter 3, near the end: “…RNA interference can employed to suppress…” is missing “be”.

Page 16, line 30 (I think, I had to count them manually…) “trigge/red”.

Page 21, Chapter 5 (conclusions). 3rd line in that paragraph: “…plant viruses. with the environment. The economic…”. 21st line: “…antiviral agents. withdraw. In this era…”. Last line: “…ensuring food safety. whole food. and environmental health.”.

Funding: Not applicable.

Starting with (Aworh 2021), in the reference list, their numeration is wrong and do not fit the number in article text.

Comments on the Quality of English Language

There were just two or three grammatical errors.

There are much more problems that are of formating/structural type.

Author Response

Dear Reviewer,

Thank you for your valuable suggestions. We have incorporated all the necessary corrections as per your recommendations. Please find our response below:

  1. Comments to Authors: Overall, not a bad summary of current research methods in plant virology. The more interesting question is what is the point of it. Such reviews are coming out annually. In 2023 alone there are at least two very similar works talking about same problems and strategies. So, my advice for the future: either start doing actual scientific experiments or, if you want to write reviews, then pick your theme better, make your article truly unique

Response: Thank you for your insightful comments on our manuscript. We appreciate your perspective on the current research methods in plant virology. Your point about the saturation of similar reviews in 2023 is duly noted. We understand the importance of offering unique contributions to the literature. Moving forward, we will consider more distinctive themes or, as suggested, focus on incorporating actual scientific experiments to enhance the novelty of our work. Your feedback is invaluable, and we will strive to improve the relevance and uniqueness of our future contributions.

  1. Comments to Authors: When presenting the problems of plant virology to the reader in the introduction, a true virologist would explain the term “quasispecies”. Most plant viruses are RNA based and are quasispecies, which means that each infected individual plant is a whole population of slightly different viruses. It is a pool of varying genomes, which, together with unparalleled mutation frequency of RNA viruses, tremendously complicates all nucleotide sequence related treatments.

Response: Thank you for your insightful comment regarding the introduction. We appreciate your suggestion to include an explanation of the term "quasispecies." In our revised version, we have ensure to provide a clear explanation of "quasispecies" to enhance the reader's understanding of the unique challenges posed by the diverse populations of viruses within individual plants.

  1. Comments to Authors: It seems that manuscript was submitted in a hurry and should be doublechecked for structural errors. No line numbers were shown in the pdf version, so it is hard to point out everything. In the abstract first word “recent” is in bold, no gap before it.

Response: We apologize for any oversight in the submission process, particularly the absence of line numbers in the PDF version. We understand the importance of a meticulous review and ensured to rectify any structural errors in the manuscript. Additionally, we appreciate your specific mention of the formatting issue in the abstract, and we have corrected the bolding and spacing of the word "recent" accordingly.

  1. Comments to Authors: In the third line of introduction, “plant viral disease” should be plural, “diseases”.

Response: In the revised version, we will correct it to "plant viral diseases" to accurately reflect the plural nature of the subject.

  1. Comments to Authors: Throughout the whole article there is often missing a gap where citations are mentioned (usually at the end of the sentence before opening parentheses. But, also sometimes between the closing parentheses after mentioning author names before citation number). Like in “…plant health(Jones 2014) [1]” or ”(Castle et al. 2009)[3]”

Response: We have carefully addressed the missing gaps in citations throughout the article, ensuring a consistent and clear presentation of references. Also formatted references according to MDPI journal.

  1. Comments to Authors: On page 7, DAS ELISA has two copies of same word: ”…sandwich (DAS) ELISA ELISA”.

Response Done

Comments to Authors: 2.2.3. chapter has different (probably 1.5) line spacing.

Response: Done

Comments to Authors: 8. 2.2.4. chapter name and the paragraph itself are in same line.

Response: Done

Comments to Authors: 2.3.1. and 2.3.2. chapter names do not have “Colon” (double dot) at their end, while most of other chapter names do.

Response: Done

Comments to Authors: 2.3.3. chapter name again not separated from its paragraph.

Response: Done

Comments to Authors: Page 15, Chapter 3, near the end: “…RNA interference can employed to suppress…” is missing “be”.

Response: Done

Comments to Authors: Page 16, line 30 (I think, I had to count them manually…) “trigge/red”.

Response: Done

Comments to Authors: Page 21, Chapter 5 (conclusions). 3rd line in that paragraph: “…plant viruses. with the environment. The economic…”. 21st line: “…antiviral agents. withdraw. In this era…”. Last line: “…ensuring food safety. whole food. and environmental health.”.

Round 2

Reviewer 1 Report (New Reviewer)

Comments and Suggestions for Authors

I am pleased with the way the manuscript was revised from a content standpoint. The only remaining issue is the referencing.  According to the journal's guidelines, the authors should use the numbering system, but they used two systems, (authors et al) and numbers. Furthermore, section 3.2 does not have any numbers. Therefore, I recommend accepting the manuscript with minor revisions to allow these references to be corrected.

Author Response

Dear Reviewer,

Thank you for your thorough review of our manuscript, and we appreciate your positive feedback on the revised content. We have addressed the remaining concerns as per your suggestion in the revised MS.

Reviewer 2 Report (New Reviewer)

Comments and Suggestions for Authors

Correct virus names according to ICTV rules: https://ictv.global/faq/names

Author Response

Dear Reviewer,

Thank you for your insightful review and constructive feedback on our manuscript. We have carefully reviewed the full MS and have corrected the virus names per ICTV rules. 

This manuscript is a resubmission of an earlier submission. The following is a list of the peer review reports and author responses from that submission.

Round 1

Reviewer 1 Report

Comments and Suggestions for Authors
1.      Would like to suggest modifying the title as “Research Advancements in RNA-Based Strategies for Disease Diagnosis and Management of Plant Viral Diseases: Overcoming Challenges and Exploring Delivery Methods”

2.      The abstract section is very general, it requires modification.

3.      The introduction section has been presented very superficially, it requires thorough revision. The authors need to mention the economic importance of plant viruses’ such as crop loss, controlling cost, impact of pesticides on agro-ecosystem sustainability and human health. Viruses as potential biotic stress infecting key crops and their losses.

4.      At present, you draw too widely from the literature and too much background text has been added to include references that have only marginal relevance to your review in section 2. Detection of Plant Viral Diseases and 8. RNA interference (RNAi): the targeted management for plant disease control.

5.      Figure 1-3 are ok and prepared in a nice way

6.      All the tables are ok, but check the title of table 1 and the referencing style of table 1 and 3. Give a bottom note for table 2 for the abbreviation

7.      I would like to strongly recommend to add a column “yield loss” in table 1 and also replacing “Effects of the disease” with “Disease symptom”

8.      For better soundness modify the “conclusion” section as “conclusion and future prospective” and give a future roadmap for plant virus control via advanced molecular technology.

Overall, the review appears to have the potential to contribute valuable updated information to the field of agricultural virus management. But, there are many miner grammatical mistakes and use of unscientific words which can be improved. Therefore, the manuscript may be considered for publication but only after addressing the above queries.

Comments on the Quality of English Language

Minor editing of the English language required

Author Response

Comments to the Author
1.      Would like to suggest modifying the title as “Research Advancements in RNA-Based Strategies for Disease Diagnosis and Management of Plant Viral Diseases: Overcoming Challenges and Exploring Delivery Methods”

Response: Changed (see title- page 1)

Comments to the Author
2.      The abstract section is very general, it requires modification.

Response: Abstract modified (page 1)

Comments to the Author
3.      The introduction section has been presented very superficially, it requires thorough revision. The authors need to mention the economic importance of plant viruses’ such as crop loss, controlling cost, and the impact of pesticides on agro-ecosystem sustainability and human health. Viruses as potential biotic stress infecting key crops and their losses.

Response: A para satisfying this comment is added, please check page 3

Comments to the Author
4.      At present, you draw too widely from the literature and too much background text has been added to include references that have only marginal relevance to your review in section 2. Detection of Plant Viral Diseases and 8. RNA interference (RNAi): the targeted management for plant disease control.

Response: corrected

Comments to the Author
5. Figures 1-3 are ok and prepared in a nice way

Response: Thanks

Comments to the Author
6.      All the tables are okay, but check the title of Table 1 and the referencing style of Tables 1 and 3. Give a bottom note for Table 2 for the abbreviation

Response: changes made

Comments to the Author
7.       I would like to strongly recommend adding a column “yield loss” in Table 1 and also replacing “Effects of the disease” with “Disease symptom”

Response:  Actually, very limited information is available on yield loss due to plant virus disease.  A systematic study is necessary in this regard; hence, very difficult to add some information on yield loss.

Comments to the Author
8.      For better soundness modify the “conclusion” section as “conclusion and future prospective” and give a future roadmap for plant virus control via advanced molecular technology. Overall, the review appears to have the potential to contribute valuable updated information to the field of agricultural virus management. However, there are many minor grammatical mistakes and the use of unscientific words which can be improved. Therefore, the manuscript may be considered for publication but only after addressing the above queries.

Response: Check page 22 for altered conclusions and future prospectives

Reviewer 2 Report

Comments and Suggestions for Authors

The manuscript by Devi and co-authors aims to review RNA-based strategies for disease diagnosis and management of plant viral diseases.

The idea of the review is of interest for the wide audience. However, the structure, the style, the form of presentation and overall quality is below publication standards of Agriculture journal. In my opinion, the MS is very superficial and poorly structured, contains errors. Below I made several comments for authors but the list is far from exhaustive.

To start from the title:

-        the first part of the sentence is not linked with the second – “strategies”…. “exploring delivery methods” – lack of logic. I assume, that authors meant delivery methods of siRNA, but it becomes clear only from the text while the title is confusing

-        “disease” is repeated twice

This title doesn’t reflect the content because in the main text not only RNA-based strategies are discussed.

The abstract:

-        the phrase “However, the use of RNA-based strategies for plant disease control faces certain limitations.” is not connected with the previous sentences, not a word about RNA-based strategies was said in the previous fragment

-        I can’t agree that “RNA interference” could be regarded as one of the “methods for detecting viruses”. It could be an approach to manage viral infection

-        “RNAi-based products, including kits and proteins” – what are these products?

The main text contains confusing, misleading or wrong phrases.

-        page 3 “Microscopy-based visualization of fluorescently labeled proteins in host plants is considered one of the most effective methods for viral diagnosis [18]” – what’ the connection between viral infection and fluorescent proteins?

-        paragraph about ELISA “sensitive and specific technique for detecting the presence of Plant Tissue Antigen (PTA) and other plant viruses” – I don’t understand what the PTA is and the sense of this phrase

-        In Dot blot assay an antigen-containing sample is blotted on the membrane and specific antibodies are applied in the incubation solution. Here again I found PTA abbreviation which stands for (Plate Trapped Antibody) in this case… An again it’s not clear what plate do the authors mean.

-        Section 2.3. mentions CRISPR/Cas technology but below there is a separate big section about this approach – the question of text structure.

-        Section 3. CRISPR/Cas technology: authors should better describe this technology, mentioning both Cas 12 and Cas 13 (for DNA or RNA targets) and explaining the detection method. As the review is for the wide audience and authors explain western- and dot-blot in details but a more complicated and less routine method utilizing Cas12 for virus detection lacks proper description and explanation.

-        Section 4. “microarray consists of small glass slides called chips” - microarray is an assay but not a chip

-        authors should consider merging sections describing “big data” methods – microarray, metagenomics, NGS, HTS as subsections into one section

-        why Tissue Blot Immunoassay section is not in the section with serological methods? In TBIA section there is a phrase “Plant virus-derived expression systems [69-73] present a multitude of benefits, encompassing elevated expression levels, swift production, scalability, safety, and economical feasibility.” which looks completely out of place there.

Table 1 is meaningless. How did the authors choose the examples for this table? the name of the table doesn’t reflect its content. Why RNAi is there as a method? What is IC-RT-PCR?

Table 2, line 1 – “Application” column contains symptoms description. Why?

Line 2 of Table 2, same column – “Tobacco for tomato virus, Nicotiana glutinosa for potato virus, and Gomphrena globose for potato X virus” – what is tomato virus or potato virus?

Starting from line 11 this table contains not detection techniques, it contains management approach

Table 3. How did the authors choose these four positions among dozens??? For review see for example: Dubrovina et al, 2019 doi:10.3390/ijms20092282; Rêgo‑Machado et al, 2023 doi.org/10.1007/s40858-022-00534-9

Incorrect terminology. E.g. table 3 “Expression of dsRNA targeting viral motor protein genes” – wrong protein name; Section 9. “VIGS is another delivery method” – this is wrong.

 Section "The impact of dsRNA on viruses control in cucurbitaceae family" doesn’t contain anything about cucurbitaceae-infecting viruses.

Next section 11 also looks out of place, its purpose is unclear.

Author Response

Comments to the Author
The manuscript by Devi and co-authors aims to review RNA-based strategies for disease diagnosis and management of plant viral diseases. The idea of the review is of interest to a wide audience. However, the structure, style, form of presentation, and overall quality are below the publication standards of the Agriculture Journal. In my opinion, the

MS is very superficial and poorly structured and contains errors. Below I made several comments for authors the list is far from exhaustive.

To start from the title:-  the first part of the sentence is not linked with the second – “strategies”…. “exploring delivery methods” –lack of logic. I assume, that the authors meant delivery methods of siRNA, but it becomes clear only from the

text while the title is confusing

-  “disease” is repeated twice

This title doesn’t reflect the content because, in the main text, not only RNA-based strategies are

Discussed.

Response: Changed title

Comments to the Author
The abstract:

-        the phrase “However, the use of RNA-based strategies for plant disease control faces certain limitations.” is not connected with the previous sentences, not a word about RNA-based strategies was said in the previous fragment

-        I can’t agree that “RNA interference” could be regarded as one of the “methods for detecting viruses”. It could be an approach to managing viral infection

-        “RNAi-based products, including kits and proteins” – what are these products?

The main text contains confusing, misleading, or wrong phrases.

-        page 3 “Microscopy-based visualization of fluorescently labeled proteins in host plants is considered one of the most effective methods for viral diagnosis [18]” – what’s the connection between viral infection and fluorescent proteins?

Response: Check page 4, added a few lines for a better understanding

Comments to the Author
-        paragraph about ELISA “sensitive and specific technique for detecting the presence of Plant Tissue Antigen (PTA) and other plant viruses” – I don’t understand what the PTA is and the sense of this phrase

Response: Page 5 Removed the previous, altered the content and provided the essential phrases.

Comments to the Author
-        In Dot blot assay an antigen-containing sample is blotted on the membrane and specific antibodies are applied in the incubation solution. Here again, I found the PTA abbreviation which stands for (Plate Trapped Antibody) in this case… Again it’s not clear what plate the authors mean.

Response: Page 6: altered the content and focused only on the Dot Blot methodology

Comments to the Author
-        Section 2.3. mentions CRISPR/Cas technology but below there is a separate big section about this approach – the question of text structure.

Response: Page 10: Changed a few phrases and satisfied the comment

Comments to the Author
-        Section 3. CRISPR/Cas technology: authors should better describe this technology, mentioning both Cas 12 and Cas 13 (for DNA or RNA targets) and explaining the detection method. The review is for a wide audience and the authors explain Western- and dot-blot in detail but a more complicated and less routine method utilizing Cas12 for virus detection lacks proper description and explanation.

Response: Page 10: Changed a few phrases

Comments to the Author
-        Section 4. “microarray consists of small glass slides called chips” - microarray is an assay but not a chip

Response: Check page 11, microarrays (altered a few points)

Comments to the Author
 -        Authors should consider merging sections describing “big data” methods – microarray, metagenomics, NGS, HTS as subsections into one section

Response: Done

Comments to the Author
-        Why ‘Tissue Blot Immunoassay’ section is not in the section with serological methods? In the TBIA section, there is a phrase “Plant virus-derived expression systems [69-73] present a multitude of benefits, encompassing elevated expression levels, swift production, scalability, safety, and economical feasibility.” which looks completely out of place there.

Response: Tissue Blot Immunoassay’ is now placed with serological section. The concern line “Plant virus-derived expression systems [69-73]” is removed from this section.

Comments to the Author
Table 1 is meaningless. How did the authors choose the examples for this table? the name of the table doesn’t reflect its content. Why RNAi is there as a method? What is IC-RT-PCR?

Response: Title of table is changed. RNAi was replaced with western blot. IC-RT-PCR is immune capture RT-PCR

Comments to the Author
Table 2, line 1 – “Application” column contains symptoms description. Why?

Response:  Table 2 was transformed as table 1 and Application part was cited with suitable examples

Comments to the Author
Line 2 of Table 2, same column – “Tobacco for tomato virus, Nicotiana glutinosa for potato virus, and Gomphrena globose for potato X virus” – what is tomato virus or potato virus?

Response: “Tobacco for tomato virus, Nicotiana glutinosa for potato virus, and Gomphrena globose for potato X virus” – is corrected in this modified version.

Comments to the Author
Starting from line 11 this table contains no detection techniques, it contains a management approach

Response: From line 11 of table 1 (previously table 2) containing different RNA based management strategies, separated as table number 3.

Comments to the Author
Table 3. How did the authors choose these four positions among dozens??? For review see for example: Dubrovina et al, 2019 doi:10.3390/ijms20092282; RêgoMachado et al, 2023 doi.org/10.1007/s40858-022-00534-9 Incorrect terminology. E.g. table 3 “Expression of dsRNA targeting viral motor protein genes” – wrong protein name; Section 9. “VIGS is another delivery method” – this is wrong.  Section " The impact of dsRNA on viruses control in cucurbitaceae family" doesn’t contain anything about cucurbitaceae-infecting viruses.

Response: 1.corrected, in table 4 (previously table 3) different citation was added for different hosts for examplifying ‘tropical application of dsRNA on host plants for plant virus management’.

2.“Expression of dsRNA targeting viral motor protein genes”..coat protein, rather than motor protein.

  1. Section 9. “VIGS is another delivery method” – this is modified
  2. Section " “The impact of dsRNA on viruses control in cucurbitaceae family&quot” –title of this section is changed

Comments to the Author
Next section 11 also looks out of place, its purpose is unclear.

Response: In section 6 (previously 11), it is necessary to exemplify a case study showing the efficacy of dsRNA

Reviewer 3 Report

Comments and Suggestions for Authors

The manuscript titled" Advancements in RNA-Based Strategies for Disease Diagnosis and Management of Plant Viral Diseases: Overcoming Challenges and Exploring Delivery Methods. Actually, the idea of the review is quite good and I see that the authors spent a lot of effort to represent their idea concerning the viral management and control. But I have some comments such as:

Title: should be changed into: RNA-Based Strategies for both Disease Diagnosis and Management of Plant Viral Diseases: Challenges and the possible Exploring Delivery Methods.

Abstract:

The conclusion at the end of the abstract is not good and should be rewritten.

Introduction

Also, the introduction should contain a general overview on the importance of virus diseases and how much it made high losses in plant yields.

Also, the plant immune against the viral infection should be included in the body of the manuscript and how many genes are involved in such mechanisms.

Conclusion

Needs more clarification and good convey is requested.

Author Response

Comments to the Author
The manuscript titled; Advancements in RNA-Based Strategies for Disease Diagnosis and Management of Plant Viral Diseases: Overcoming Challenges and Exploring Delivery Methods. The idea of the review is quite good and I see that the authors spent a lot of effort to represent their idea concerning viral management and control. But I have some comments such as:

Title: should be changed into RNA-Based Strategies for both Disease Diagnosis and Management of Plant Viral Diseases: Challenges and the Possible Exploring Delivery Methods.

Response: Page 1: Title changed

Comments to the Author
Abstract:

The conclusion at the end of the abstract is not good and should be rewritten.

Response:  Page 1 abstract altered

Comments to the Author
Introduction

Also, the introduction should contain a general overview of the importance of virus diseases and how much they caused high losses in plant yields.

Response: Page 2: added a few phrases

Comments to the Author
Also, the plant immunity against the viral infection should be included in the body of the manuscript and how many genes are involved in such mechanisms.

Response:

Here, in this manuscript, we are discussing an RNAi-mediated defense system for plant virus management and its mechanism (in section 3). We are least interested about plant immunity and related genes.

Comments to the Author
Conclusion: Needs more clarification and a good convey is requested.

Response:  modification done

Round 2

Reviewer 2 Report

Comments and Suggestions for Authors

Authors made some effort to reply to my comments, however, they didn't answer all the questions.

Abstract was rewritten but not improved because now it doesn't reflect the MS content and devoted only to RNAi, nothing is said there about diagnostic tools. 

Overall the manuscript is still superficial.

Literature cited in the MS is often irrelevant.

Numerous cases of improper terminology used are present. Also there are mistakes and misleading phrases.

Author Response

Dear Reviewer,

1. Comments to Authors: The authors made some effort to reply to my comments, however, they didn't answer all the questions.

Response: We appreciate your feedback and thank you for taking the time to review our manuscript. We have carefully considered your comments and made every effort to address them in our revised manuscript. 

2. Comments to Authors: Abstract was rewritten but not improved because now it doesn't reflect the MS content and devoted only to RNAi, nothing is said there about diagnostic tools. 

Response: We have made the necessary revisions to address this issue and ensure that the abstract provides a balanced and comprehensive overview of both RNAi and diagnostic tools.

3. Comments to Authors: Overall the manuscript is still superficial.

Response: We have addressed your all concerns, and conducted a thorough review of the manuscript, ensuring that we provide more in-depth analysis, discussion, and content to improve the overall quality of the paper.

4. Comments to Authors: Literature cited in the MS is often irrelevant.

Response: Thank you for your feedback. We apologize for any inaccuracies or irrelevance in our references. We have reviewed the manuscript and ensured that all references are appropriately matched with the content.

5. Comments to Authors: Numerous cases of improper terminology used are present. Also, there are mistakes and misleading phrases

Response: Thank you for bringing this to our attention. We have conducted a thorough review of the manuscript, addressed any errors, and enhanced the clarity of the content.